# GENIE ENVISIONER: A UNIFIED WORLD FOUNDATION PLATFORM FOR ROBOTIC MANIPULATION

**Yue Liao**[*]    **Pengfei Zhou**[*]    **Siyuan Huang**[*]    **Donglin Yang**    **Shengcong Chen**
**Yuxin Jiang**    **Yue Hu**    **Jingbin Cai**    **Si Liu**    **Jianlan Luo**    **Liliang Chen**[†]
**Shuicheng Yan**[◇]    **Maoqing Yao**[◇]    **Guanghui Ren**[†◇]

AgiBot Genie Team    LV-NUS Lab    BUAA

https://genie-envisioner.github.io

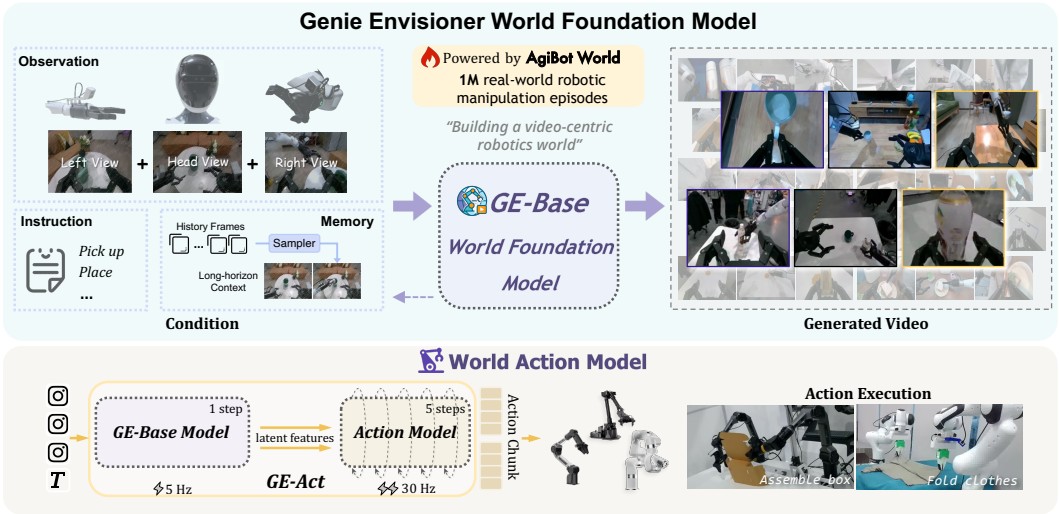

Figure 1: **Overview of the Genie Envisioner World Foundation Model.** Genie Envisioner (GE) is a unified world foundation model that integrates robotic world generation and manipulation policy learning within a single video-generative framework. At its core is GE-Base, a large-scale video world model that encodes the spatial, temporal, and semantic structure of robotic interactions. Building on this foundation, GE-Act functions as a world action model that derives instruction-conditioned policies from the embodied video space to enable embodied control.

## ABSTRACT

We introduce Genie Envisioner (GE), a unified world foundation platform for robotic manipulation that jointly learns visual representations and action policies within a single video-generative framework. At its core, GE-Base is a large-scale instruction-conditioned video diffusion model that captures the spatial, temporal, and semantic dynamics of real-world robotic interactions in a structured latent space. Building on this foundation, GE-Act employs a lightweight flow-matching decoder to map latent representations into executable action trajectories, enabling precise and generalizable policy inference across diverse embodiments with minimal supervision. Trained on over 1 million manipulation episodes, GE supports both short- and long-horizon tasks, and generalizes across embodiments.

## 1 INTRODUCTION

Embodied agents that sense, reason, and act in the physical world represent the next frontier of AI systems. At its core, a fundamental challenge remains: developing scalable and robust robotic manipulation capabilities - the ability to purposefully interact with and control the physical environment via selective contact (Mason, 2001). Progress in this domain has spanned analytic methods (Stilman,

---

[*] Equal Contribution. [†] Project Leader. [◇] Corresponding Author.

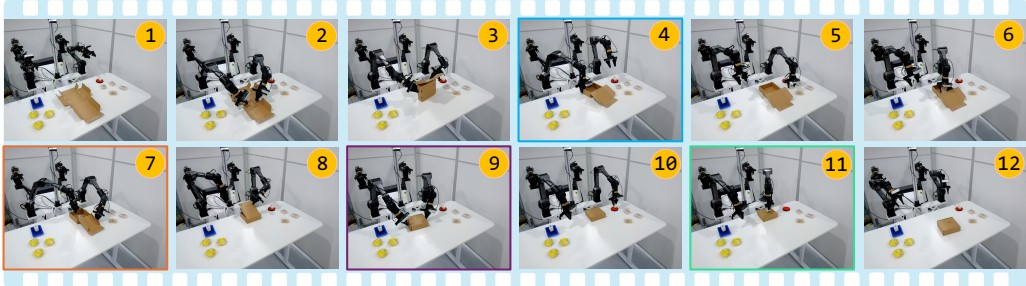

*"Yellow candy requires a blue stamp, white candy requires a red stamp. Fold a box, place the appropriate candy inside, seal the box, and apply the correct stamp based on the candy type."*

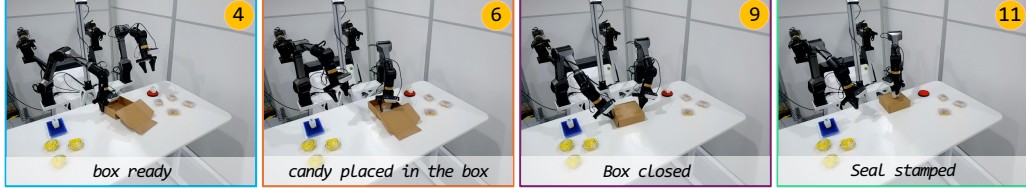

Figure 2: **Real-world demonstration of GE-Act on a novel robot embodiment, Agilex Cobot Magic, unseen during pretraining.** With only one hour of embodiment- and task-specific teleoperation data for post-training, GE-Act successfully executes a complex manipulation task involving fine-grained control of deformable objects and memory-based decision making. Given a general packaging rule, the robot is required to complete the packing process for each item accordingly. Here, we showcase the detailed execution of the first packing cycle. The robot first folds a deformable box, places a target object inside based on instruction, and closes the lid, *rendering the object no longer visible*. It then correctly selects and applies the appropriate stamp, matching the object type, relying solely on internal memory. This showcases GE's generalization to new embodiments, its precise handling of deformable materials, and its ability to retain task-relevant memory across steps. .

2007; Berenson et al., 2009), model-based frameworks (Ebert et al., 2018; Janner et al., 2019; Nagabandi et al., 2020), and more recently, data-driven approaches that learn manipulation policies from large-scale datasets (Brohan et al., 2023; Kim et al., 2024; Black et al., 2024; Bu et al., 2025b).

Although vision–language–action (VLA) imitation learning has achieved notable progress in generalization and task performance, it remains fundamentally constrained by its reliance on language-centric representations. Mainstream VLAs compress visual observations into low-bandwidth semantic embeddings, which, while effective for high-level understanding, fail to explicitly encode future dynamics and therefore fall short in supporting fine-grained motor control. Attempts to combine VLMs with diffusion-based policies (Black et al., 2024) often destabilize training, as continuous action losses overshadow linguistic objectives and distort pretrained weights.

Recently, video-centric world models (Agarwal et al., 2025) have emerged as a powerful alternative, shifting the dominant paradigm from *"vision → language"* to *"language → future video."* This shift enables models to represent motion dynamics, contact evolution, and fine-grained perceptual cues that are inherently abstracted away in purely semantic embeddings, making video-centric representations a more suitable foundation for manipulation. Motivated by this direction, several recent works (Wu et al., 2023a; Cheang et al., 2024; Hu et al., 2024b; Liang et al., 2025) explore policy architectures that derive actions directly from video diffusion latents or even decode actions directly from pixel-space video predictions. Although promising, these approaches still exhibit important limitations. Most rely on single-view video generation, which is misaligned with the multi-view egocentric perception typical of real robots. Moreover, both pixel-level action decoding and latent-based serial video-to-action pipelines incur substantial inference latency and therefore require aggressive latent compression to remain tractable. Such compression leads to the loss of critical fine-grained spatial and contact cues, ultimately constraining the precision required for real-world manipulation.

To this end, we introduce **Genie Envisioner (GE)**, a world foundation model for robotic manipulation that unifies ego-centric visual world modeling and policy learning within a single closed-loop generative architecture (Figure 1). At its core is GE-Base, an instruction-conditioned multi-view video diffusion model that explicitly predicts future head-view and wrist-view observations, enforcing

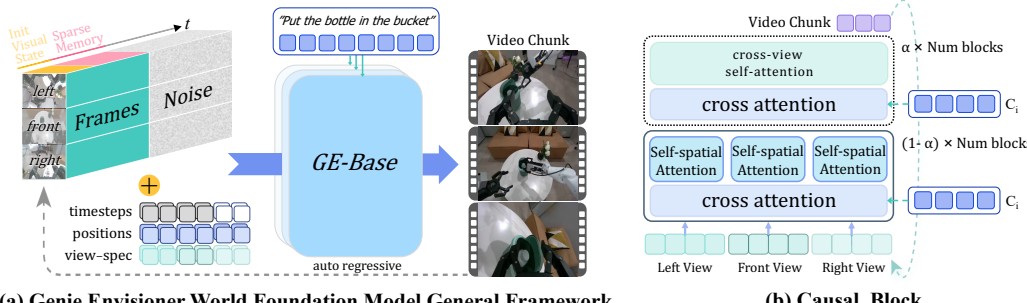

**(a) Genie Envisioner World Foundation Model General Framework**   **(b) Causal Block**

Figure 3: **Overview of the GE-Base.** (a) An illustration of the autoregressive video generation process. Given multi-view visual conditions, the model generates the next multi-view video chunk conditioned on a language instruction. (b) A dedicated causal block facilitates information exchange across different views, ensuring spatial consistency during multi-view video chunk generation.

cross-view consistency across perspectives. This multi-view formulation aligns the generative process with real robot embodiment and constructs a high-fidelity latent visual space rich in spatial, temporal, and semantic structure. GE-Base is trained on 3,000 hours of paired video–language demonstrations (about 1 million episodes) from AgiBot-World-Beta (Bu et al., 2025a), enabling it to autoregressively simulate manipulation trajectories and provide dense, multi-scale latent representations.

Building on this representation, we introduce **GE-Act**, a lightweight parallel world action module that is block-wise aligned with GE-Base. Unlike prior serial video-to-action pipelines that compress video latents before policy decoding, GE-Act operates in parallel with the video generator and directly accesses the full-resolution, multi-scale latent features at each DiT block via cross-attention. This parallel, block-aligned design preserves the fine-grained geometry, motion cues, and contact details that are typically lost in compressed VLA pipelines, enabling significantly more accurate manipulation. To further meet real-time control demands, GE employs an asynchronous inference strategy: GE-Base updates its heavy video diffusion branch using a single denoising step at a low frequency, while GE-Act updates at a higher frequency with multi-step action denoising. This slow–fast scheduling provides dense action outputs and sparse video refreshes, dramatically reducing computation without sacrificing control fidelity.

We comprehensively evaluate GE in both embodied video generation and policy learning across diverse real-world manipulation tasks. GE-Act achieves low-latency end-to-end control, generating *54-step torque trajectories within 200 ms on a commodity GPU*. It executes tasks precisely on the in-domain AgiBot G1 and generalizes effectively to novel systems such as Dual Franka and Agilex Cobot Magic with only one hour of teleoperated demonstrations, surpassing task-specific baselines (Black et al., 2024; Bjorck et al., 2025; Bu et al., 2025b). GE-Act also performs reliably across industrial and household scenarios. Beyond short-horizon tasks, its visual world modeling supports long-horizon, memory-intensive sequences (Figure 2). We further assess GE-Base on EWMBench (Yue et al., 2025), benchmarking its embodied video generation against state-of-the-art models. GE-Base consistently outperforms alternatives, highlighting its role as the foundation of GE.

Together, these contributions position Genie Envisioner as a practical, scalable foundation for real-world manipulation, facilitating downstream research.

## 2 GE-BASE: WORLD FOUNDATION MODEL

In this section, we introduce **GE-Base**, the core of Genie Envisioner. It extends general video generation into *an embodied predictive representation* that anticipates robot–environment interactions from task instructions and initial observations. Unlike generic video models, GE-Base is a fully egocentric multi-view generator, synthesizing head and wrist perspectives with enforced cross-view consistency to build coherent embodied spaces. It adopts a video DiT with robotic-adaptive pretraining, transferring knowledge from large-scale video corpora to the embodied domain.

## 2.1 BASIC ARCHITECTURE

To model long-horizon robotic manipulation, we adopt an autoregressive chunk-wise video generation framework. At autoregressive step $t$, the world model $\mathcal{W}$ predicts a chunk of $N$ consecutive frames, denoted $\mathbf{x}_{1:N}^t$. The prediction is conditioned on the initial multi-view observation $\mathbf{x}_0$, the instruction embedding $\mathcal{T}(q)$, and a long-term sparse memory $\mathbf{m}_{0:t-1}$. The memory $\mathbf{m}$ is constructed by sparsely sampling keyframes from previously generated chunks $\{\mathbf{x}_{1:N}^k\}_{k=0}^{t-1}$, enabling the model to retain extended temporal context. The autoregressive generation process is expressed as

$$\mathbf{x}_{1:N}^t = \mathcal{W}(\mathbf{x}_0, \mathbf{m}_{0:t-1}, \mathcal{T}(q)).$$

As illustrated in Figure 3, at each step $\mathcal{W}$ receives synchronized observations from $V$ onboard cameras, such as a head-mounted view and two wrist-mounted views. Each view $i$ provides its initial observation $\mathbf{x}_0^i$, its sparse memory frames $\mathbf{m}_{0:t-1}^i$, and a view-specific noise map $\mathbf{z}^i$. These inputs are encoded by a shared video encoder $\mathcal{E}$: $\mathbf{v}_0^i = \mathcal{E}(\mathbf{x}_0^i)$, and $\mathbf{v}_\mathbf{m}^i = \mathcal{E}(\mathbf{m}_{0:t-1}^i)$, and each token is enriched with a 3D rotary positional embedding and a learnable view embedding:

$$\tilde{\mathbf{v}}^i = \text{RoPE}(t, h, w) + \mathbf{v}^i + \mathbf{e}_{\text{view}}^i.$$

The final per-view input visual tokens are $\mathbf{u}^i = \left[\tilde{\mathbf{v}}_0^i \parallel \tilde{\mathbf{v}}_\mathbf{m}^i \parallel \mathbf{z}^i\right]$, and tokens from all views are concatenated and passed into the DiT backbone, along with timestep and instruction embedding.

To ensure consistency across viewpoints, we introduce cross-view attention in selected transformer blocks. Tokens from all views are temporarily merged into a single latent sequence, allowing each view to attend to others and promoting coherent geometry and motion across perspectives. For efficiency, only a subset of blocks use cross-view attention, while the rest process views independently, achieving a balanced trade-off between multi-view coherence and computational cost.

To balance efficiency and modeling capacity, we employ the compact LTX-Video 2B (HaCohen et al., 2024) architecture as the backbone for $\mathcal{W}$.

The model is trained with a latent diffusion objective. Given VAE latents $\mathbf{l}$ of the target video chunk and a noisy latent $\tilde{\mathbf{l}} = (1 - \sigma_\tau)\mathbf{l} + \sigma_\tau\boldsymbol{\epsilon}$, generated using Gaussian noise $\boldsymbol{\epsilon} \sim \mathcal{N}(0, I)$ at timestep $\tau$, the world model predicts the denoising velocity $\mathbf{v}_\theta$. Supervision is applied only to future frames via a conditioning mask $\mathbf{M}$, giving the training loss

$$\mathcal{L}_{\text{video}} = w(\tau) \left\|\left(\mathbf{v}_\theta - (\boldsymbol{\epsilon} - \mathbf{l})\right) \odot (1 - \mathbf{M})\right\|_2^2.$$

This unified modeling paradigm enables $\mathcal{W}$ to jointly capture spatial layouts, temporal dynamics, and semantic intent, yielding coherent and controllable predictions of embodied robotic behavior.

## 2.2 WORLD MODEL PRE-TRAINING

A core challenge in robotic world models is adapting general video generation to the structured dynamics and semantics of embodied interaction. To address this, we design a multi-stage pretraining framework that progressively aligns spatiotemporal representations with the distribution of real-world robot behavior. Our data pipeline incorporates sparse memory frames randomly sampled from prior history, which increase prediction difficulty and enhance robustness to temporal variation, ultimately improving generalization across diverse manipulation scenarios.

**Data Curation.** We pretrain on the AgiBot-World-Beta dataset (Bu et al., 2025a), which contains about 1M high-quality dual-arm manipulation episodes. Each trajectory is annotated with language instructions, multi-view observations, and action policies, spanning diverse tasks, objects, and environments. For video-based modeling, we extract synchronized streams from three calibrated cameras and ensure semantic alignment with paired instructions, yielding high-quality text–video pairs that reflect coherent manipulation behaviors.

**Stage I: Multi-Resolution Temporal Adaptation (GE-Base-MR).** To bridge generic video representation learning and robotic motion dynamics, we pretrain GE-Base on 57-frame clips sampled at 3–30 Hz. Each sample includes four sparse memory frames to enhance temporal diversity. Clips are compressed into an 8-frame latent space via a pretrained video VAE and optimized with a denoising objective. This exposes GE-Base-MR to diverse motion speeds and partial observations, enabling spatiotemporal representations invariant to sampling rates.

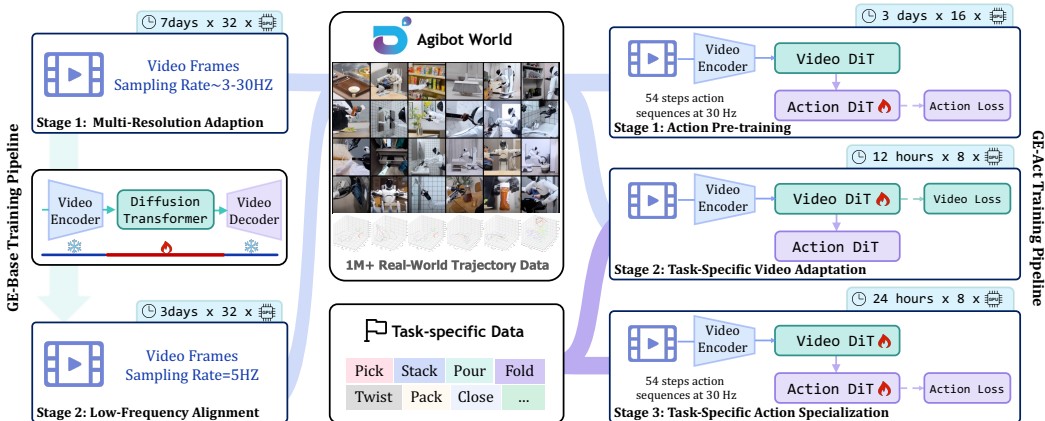

Figure 4: **Training Pipelines of GE-Base and GE-Act.** *Left:* GE-Base is pretrained on AgiBot-World-Beta and training includes domain adaptation with high-frame-rate sampling for robustness, followed by low-frame-rate fine-tuning to align temporal resolution with downstream policy learning. *Right:* GE-Act is derived from GE-Base through a three-stage pipeline using text–video–policy triplets. It first performs action-space pretraining, then applies two-stage task adaptation on task data.

**Stage II: Low-Frequency Policy Alignment (GE-Base-LF).** To align with the temporal abstraction of downstream control, we fine-tune GE-Base-MR on 9-frame clips sampled at 5 Hz, with four additional memory frames as context. Sequences are encoded into two latent frames by a frozen video encoder, while only the generation components are updated. GE-Base-LF is optimized to capture semantically meaningful transitions under sparse sampling, enabling reliable video feedback at the granularity of action steps. This provides the foundation for subsequent action model pretraining.

## 2.3 ROBOTIC MANIPULATION VIDEO GENERATION VIA GE-BASE

We generate dual-arm robotic manipulation videos using GE-Base. Generation follows an autoregressive scheme: each step produces a video chunk conditioned on the initial observation, memory frames, and language instruction, and the process iterates until the instructed task is fully executed. During inference, memory frames are uniformly sampled at fixed intervals from prior chunks to ensure stable temporal dynamics and consistent predictions.

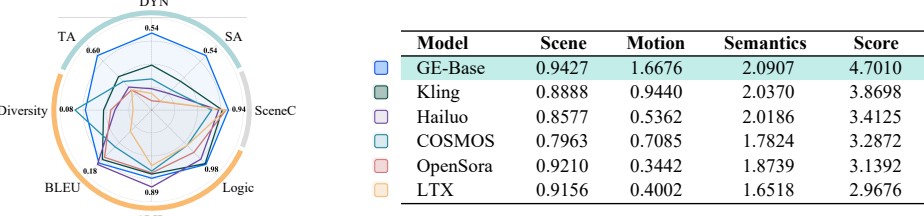

| Model | Scene | Motion | Semantics | Score |
|---|---|---|---|---|
| GE-Base | 0.9427 | 1.6676 | 2.0907 | 4.7010 |
| Kling | 0.8888 | 0.9440 | 2.0370 | 3.8698 |
| Hailuo | 0.8577 | 0.5362 | 2.0186 | 3.4125 |
| COSMOS | 0.7963 | 0.7085 | 1.7824 | 3.2872 |
| OpenSora | 0.9210 | 0.3442 | 1.8739 | 3.1392 |
| LTX | 0.9156 | 0.4002 | 1.6518 | 2.9676 |

(a) Fine-Grained Evaluation of Video Generation Models     (b) Aggregated Evaluation Across Hierarchical Levels

Figure 5: **Comparison of GE-Base and Video World Models.** We compare GE-Base with SOTA video generation models on EWMBench. Evaluation spans scene, motion, and semantic levels.

We evaluate this pipeline on real-world dual-arm manipulation generation on EWMBench (Yue et al., 2025), comparing GE-Base against seven SOTA video generation models (Zheng et al., 2024; Kuaishou, 2025; MiniMax, 2024; HaCohen et al., 2024; Agarwal et al., 2025) under a standardized text-and-image-to-video setting. As shown in Figure 5, GE-Base consistently outperforms baselines across multiple dimensions, excelling in temporal alignment and dynamic consistency, two core requirements for generating action-plausible and temporally stable robotic behaviors. While its motion semantics are comparable to generic models, GE-Base achieves substantially higher control fidelity, producing more precise and reliable task executions. This advantage is attributed to pretraining on large-scale robotic data, enabling better capture of task-relevant spatiotemporal dynamics.

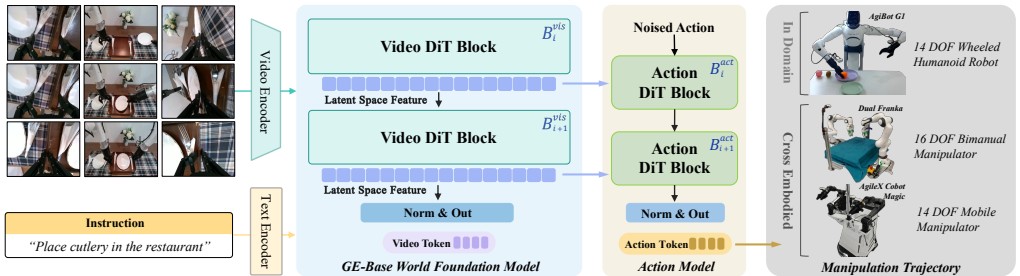

Figure 6: **Overview of the GE-Act World Action Model.** GE-Act extends GE-Base with a parallel action branch that converts visual latents into structured policy trajectories. It follows the same block design as GE-Base with reduced hidden dimensions for efficiency. Visual features are integrated through cross-attention for semantic grounding, and final actions are produced via a diffusion-based flow-matching pipeline that refines noisy predictions into coherent trajectories. By default, action blocks cross-attend to multi-scale video features; at inference, denoised visual latents are cached with few denoising steps (M) and reused, whereas the action branch denoises for N > M steps.

## 3 GE-ACT: WORLD ACTION MODEL

Bridging high-level world modeling and low-level control is critical for deploying world foundation models in robotics. We present **GE-Act**, a plug-and-play parallel module that augments GE-Base with a lightweight 160M-parameter autoregressive decoder. GE-Act is block-wise aligned with GE-Base, enabling it to parallelly access the full hierarchy of multi-scale latent visual features produced throughout the video diffusion backbone. This parallel design preserves rich spatial information across multi-view observations and allows GE-Act to consume high-resolution latent representations without waiting for explicit video decoding. Conditioned on both multi-view perception and language instructions, GE-Act directly maps these multi-scale multimodal latents into structured action policies, enabling instruction-following without explicit video generation. This tight block-wise coupling between perception and control provides a scalable and efficient solution for real-time manipulation.

### 3.1 BASIC ARCHITECTURE

As shown in Figure 6, GE-Act mirrors the DiT depth of GE-Base while adopting a reduced hidden dimension for computational efficiency. At autoregressive step $t$, GE-Base processes visual tokens derived from the initial observation $\mathbf{x}_0$, the sparse memory $\mathbf{m}_{0:t-1}$, and the language embedding $\mathcal{T}(q)$, producing multi-scale latent features across its DiT blocks:

$$\mathbf{v}_i = \mathcal{B}_i^{\text{vis}}\big(\mathbf{v}_{\text{in}},\ \mathcal{T}(q)\big),$$

where $\mathbf{v}_{\text{in}}$ denotes the fused visual tokens from $\mathbf{x}_0$ and $\mathbf{m}_{0:t-1}$. It notes that GE-Act uses a different form of memory is sampled directly from the robot's historical visual observations rather than from previously generated frames (as in GE-Base). This design ensures that GE-Act conditions on accurate, real-world perceptual history when producing actions.

In parallel, GE-Act operates entirely in the latent action space. It initializes action tokens $\mathbf{z}_{\text{act}}$ with Gaussian noise and updates them through a sequence of action-specific DiT blocks that attend to the corresponding multi-scale visual features:

$$\mathbf{a}_i = \mathcal{B}_i^{\text{act}}\big(\mathbf{z}_{\text{act}},\ \text{CrossAttn}\big(\mathbf{z}_{\text{act}},\ \mathbf{v}_i\big)\big),$$

where $\mathcal{B}_i^{\text{act}}$ is the $i$-th action DiT block. Because GE-Act is block-wise aligned with GE-Base, it receives multi-scale visual representations at matching depths, rather than relying on the final-layer visual latents as in traditional VLA pipelines. This block-level alignment allows GE-Act to exploit high-resolution spatial cues and cross-view correspondences throughout the generation process.

Following the latent diffusion objective used in GE-Base, GE-Act trains its policy decoder with a noise-conditioned velocity-matching loss. Given ground-truth actions $\mathbf{u}$, sampled noise $\boldsymbol{\epsilon}$, and timestep $\tau$ with noise level $\sigma_\tau$, we form noisy actions

$$\tilde{\mathbf{u}} = (1 - \sigma_\tau)\mathbf{u} + \sigma_\tau \boldsymbol{\epsilon},$$

and GE-Act predicts a denoising velocity $\mathbf{v}_\theta^{\text{act}}$. The supervision target is the diffusion velocity $\boldsymbol{\epsilon} - \mathbf{u}$, giving the loss

$$\mathcal{L}_{\text{act}} = w(\tau) \left\| \mathbf{v}_\theta^{\text{act}} - \left( \boldsymbol{\epsilon} - \mathbf{u} \right) \right\|_2^2,$$

where $w(\tau)$ is the timestep weighting function used in modern diffusion models. This objective mirrors the GE-Base training formulation and enables GE-Act to learn smooth, temporally coherent action trajectories directly in the latent space.

### 3.2 TRAINING PROCEDURE

We adopt a two-stage training paradigm inspired by standard VLA manipulation frameworks, consisting of task-agnostic pretraining followed by task-specific adaptation.

**Pre-training.** We pretrain the action model on AgiBot-World-Beta to adapt GE-Base for policy learning. The world model is initialized from GE-Base-LF with frozen parameters, while only the action decoder is updated. To reduce cost, video generation is disabled; instead, four low-rate (5 Hz) visual memory frames are used, and the model predicts 54-step high-frequency (30 Hz) actions.

**Task-specific adaptation.** For downstream tasks, we adopt a two-stage fine-tuning pipeline of video adaptation and action specialization. In the first stage, only video generation components are updated on a composite dataset of AgiBot-World and task-specific data, with the latter upweighted for alignment. In the second stage, the full model is fine-tuned on task-specific data to capture fine-grained dynamics, following the same sampling strategy.

### 3.3 ASYNCHRONOUS INFERENCE

To improve the efficiency of our parallel GE-Act architecture, we introduce **Slow-Fast Asynchronous Inference**e, which combines two independent forms of asynchrony. The first component is *diffusion-step asynchrony*: the video DiT performs only a single diffusion step each time it refreshes the visual latent features, while the action decoder continues to apply multi-step denoising to preserve the accuracy and stability required for fine-grained control. This design substantially reduces the computational burden of the heavy video module, yet still provides visual representations that are sufficiently informative for policy decoding. The second component is *frequency asynchrony*: the video diffusion branch operates at a low update frequency, whereas the action decoder runs at a higher frequency to support responsive action prediction. These two schedules correspond to our inference modes, where GE-Act Slow updates both branches at the same rate, and GE-Act Fast updates video at 5 Hz and actions at 30 Hz. This slow–fast scheme enables sparse video updates and dense action generation, allocating computation where it is most effective and allowing GE-Act Fast to support a 54-step prediction window and the execution of 30 action steps within 200 ms on an RTX 4090. During training, hidden states are initialized with Gaussian noise to avoid repeatedly loading full videos, and during deployment, the combination of single-step video denoising and low-frequency video updates provides lightweight but timely perceptual inputs that enable GE-Act to integrate video world modeling with action execution in real time.

### 3.4 ACTION PLANNING VIA GE-ACT ON AGIBOT G1

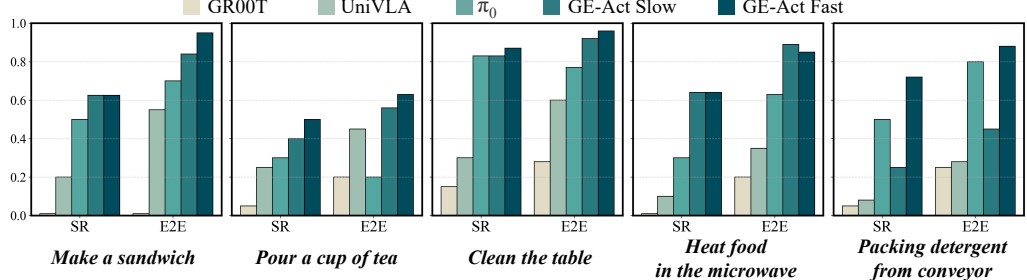

Figure 7: **Comparison of Task-Specific Real-World Robotic Manipulation Performance on the AgiBot G1 Platform.** We compare GE-Act with state-of-the-art VLA baselines across multiple real-world dual-arm robotic tasks, using two evaluation metrics to assess performance.

To rigorously evaluate our approach in real-world robotic manipulation, we conduct experiments across five tasks, each targeting different aspects of control precision, task complexity, and generaliza-

Figure 8: Comparison of Task-Specific Manipulation Performance on Various Embodiments.

tion. Spanning both household and industrial settings (details in Appendix A.1), these tasks provide a comprehensive benchmark for assessing instruction-conditioned control and closed-loop execution.

**Evaluation Protocols.** We evaluate performance using two metrics: Step-wise Success Rate (SR) and End-to-End Success Rate (E2E). SR measures the ratio of successful sub-steps to total sub-steps, offering fine-grained insight into partial task completion. E2E considers only the final task outcome, allowing recovery from intermediate failures, thus better reflecting real-world deployment.

**Performance Comparison on the AgiBot G1.** We benchmark GE-Act against three leading VLA models, $\pi_0$, UniVLA (Bu et al., 2025b), and GR00T N1 (Bjorck et al., 2025), on AgiBot G1 under identical protocols and fine-tuning data. As shown in Figure 7, GE-Act consistently surpasses baselines on SR and E2E across diverse tasks, benefiting from GE-Base pretraining that provides strong spatiotemporal priors. Qualitative results are provided in Appendix A.4.

We further compare two inference modes: synchronized and asynchronous. The asynchronous mode achieves comparable or superior results, particularly in latency-sensitive tasks, *e.g.*, dynamic tracking, and significantly outperforms the standard mode on short-horizon tasks (*e.g.*, packing detergent).

## 4 CROSS-EMBODIMENT GENERALIZATION WITH GENIE ENVISIONER

Beyond in-domain evaluation on AgiBot G1, we test GE's cross-embodiment generalization on two widely used platforms, the Franka arm and Agilex Cobot Magic, configured for consistency with our dual-arm framework. Direct deployment is infeasible due to differences in embodiment and action space, so we adopt a few-shot protocol: collecting a small set of teleoperated demonstrations to fine-tune both GE and GE-Act. In addition to standard tasks, we include challenging deformable-object manipulation, such as *"cloth folding"* and *"box folding"* to assess transferability and robustness.

**Few-shot Adaptation.** We use a two-stage adaptation strategy. First, the video DiT is adapted to the new embodiment via text-video demonstrations, while CLIP and video encoders remain frozen to preserve pretrained priors. Second, a new action DiT is trained from scratch on task-specific trajectories, reusing the GE-Base but learning a task-specific action head tailored to the new platform. This pipeline enables effective transfer of perceptual and motor capabilities with minimal supervision.

### 4.1 GENERALIZATION TO DUAL FRANKA EMBODIMENT

We evaluate GE-Act for cross-embodiment generalization on the Dual Franka platform by adapting it with 250 teleoperated episodes (1 hour) on cloth folding, collected via a space-mouse interface. Following the same protocol as Agilex Cobot, we fine-tune VLA baselines (Bjorck et al., 2025; Black et al., 2024; Bu et al., 2025b) on the same dataset. As shown in Figure 8 (a), GE-Act consistently outperforms task-specific baselines on real-world execution. Notably, despite $\pi_0$ and GR00T N1 being trained with large-scale Franka data, GE-Act achieves superior performance.

### 4.2 GENERALIZATION TO AGILEX COBOT MAGIC EMBODIMENT

We evaluate the generalization capability of GE on the Agilex Cobot Magic using two tasks: *"box folding"* and *"cloth folding"*. For each task, we collect 250 high-quality teleoperated demonstrations (approximately 1 hour of data) using the Aloha-based teleoperation system (Fu et al., 2024). These demonstrations serve as the adaptation dataset to fine-tune both GE-Base and GE-Act.

*"Folding the box"*

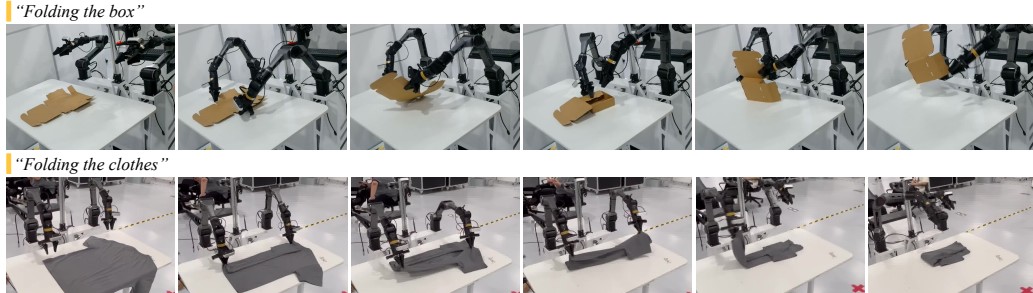

*"Folding the clothes"*

Figure 9: **Visualization of Real-World Demonstrations with GE-Act on Agilex Cobot Magic Platform.** This shows GE-Act adapted to a novel Agilex Cobot Magic embodiment, performing real-world robotic manipulation tasks, including cloth-folding and box-folding.

As shown in Figure 8 (b), we compare GE-Act with three SOTA VLA models (Bjorck et al., 2025; Black et al., 2024; Bu et al., 2025b), all fine-tuned on the same dataset. GE-Act consistently outperforms all baselines. While UniVLA and GR00T N1 perform well on simple tasks such as pick-and-place, they fail on complex, fine-grained tasks, achieving $0\%$ success. $\pi_0$ shows stronger results on deformable object manipulation, yet GE-Act significantly surpasses it in complex deformable scenarios. This advantage stems from the large-scale pretraining of GE-Base, which provides robust spatiotemporal priors, enabling superior generalization across diverse platforms and tasks. Furthermore, as shown in Figure 9, real-world executions of cloth and box folding demonstrate that the adapted GE-Act completes tasks with high precision and reliability on a novel embodiment.

### 4.3 GENERALIZATION TO ROBOTWIN

We further evaluate the cross-embodiment generalization on the dual-arm simulator RoboTwin (Chen et al., 2025). We adopt an all-in-one strategy, jointly fine-tuning GE-Act on four tasks using 200 demonstrations (50 per task), and directly evaluating this unified model across all tasks. In contrast, baseline methods (Black et al., 2024; Bu et al., 2025a) perform task-specific adaptation. As shown in Figure 8 (c), GE-Act achieves better performance than $\pi_0$ and GO-1 (Bu et al., 2025a) on three of the four tasks, despite not using a one-task-one-model setting, and is only slightly behind VLA methods on lift pot. This minor gap may be attributed to task interference introduced by joint training.

## 5 ANALYSIS

To systematically analyze our GE model, we conduct real-world robotic manipulation experiments on the AgiBot-G1 platform. We select a stable and controllable task, *"grasping a red cylinder from the table and placing it into a paper cup with fixed positions"*, using a dataset of 305 demonstrations. All models are trained for 40,000 steps under the same protocol. Our analysis focuses on the role of pretraining in action policy prediction, comparing general video pretraining with in-domain embodied pretraining (AgiBot-World-Beta). As shown in Table 1, training from scratch or adapting from a general video model such as LTX-Video yields near-zero success.

In contrast, in-domain pretraining achieves 64 SR and 81 E2E, which further improve to 76% and 89% when combined with general video pretraining. We further validate the effectiveness of incorporating robot state as input, which yields additional performance gains. However, when applied directly to general video-pretrained models, the inclusion of state information reduces performance due to short-cut learning effects. These results demonstrate that the GE-Base pretrained world model offers strong representations and serves as a solid foundation for action policy prediction.

Table 1: Experimental Analysis of Pre-training. 'S' denotes inclusion of robot state; 'VidAW' indicates initialization from GE-Base, 'VidAda' indicates task-specific video adaptation statge mentioned in 3.2.

| VidAW | VidAda | E2E | | SR | |
|:---:|:---:|:---:|:---:|:---:|:---:|
| | | w/ $\mathcal{S}$ | w/o $\mathcal{S}$ | w/ $\mathcal{S}$ | w/o $\mathcal{S}$ |
| ✗ | ✗ | 0.15 | 0.30 | 0.05 | 0.11 |
| ✗ | ✓ | 0 | 0.05 | 0 | 0 |
| ✓ | ✗ | 0.81 | 0.49 | 0.64 | 0.26 |
| ✓ | ✓ | **0.89** | 0.37 | **0.76** | 0.37 |

## 6 RELATED WORK

**Vision–Language–Action Policies.** Instruction-conditioned policies transfer from vision–language pretraining and are fine-tuned via imitation (Driess et al., 2023; Brohan et al., 2023; Kim et al., 2024; Black et al., 2024), or use VLMs as encoders/planners (Nair et al., 2022; Ahn et al., 2022; Huang et al., 2023). These approaches excel at semantic grounding but lack explicit generative rollouts for dynamics or counterfactual reasoning. Our framework conditions a generative world model on language, preserving a direct path to control while enabling predictive simulation.

**Embodied Video World Models.** World models provide predictive structure for perception and planning (Sutton & Barto, 1981; Chatila & Laumond, 1985), evolving from analytic formulations (Murray et al., 2017) to neural variants in pixel and latent spaces (Ha & Schmidhuber, 2018; Finn et al., 2016; Ebert et al., 2018; Hafner et al., 2019; Wu et al., 2023b; Hu et al., 2024a). Recent video-centric approaches shift toward language→future video (Bruce et al., 2024; Agarwal et al., 2025; Russell et al., 2025; Jang et al., 2025) but typically emphasize single-view synthesis without closed-loop control. GE-Base instead targets egocentric multi-view generation with cross-view consistency and sparse-memory autoregression for long-horizon coherence, yielding an embodied latent space aligned with manipulation.

**Coupling World Models to Action.** Recent manipulation systems that use video generation or world models can be grouped by how perception is coupled to control: (A) *Video-as-policy backbones*: a video model is used purely as a visual–temporal encoder whose compressed latents are fed into a lightweight action head. Methods in this family (Hu et al., 2024b; Li et al., 2025; Liang et al., 2025; Wen et al., 2024) typically adopt a serial video→action pipeline, which risks losing fine-grained motion and contact cues due to heavy latent compression. (B) *Unified video–action generators*: these models jointly generate video and actions within a single network (Cheang et al., 2024; Wu et al., 2023a; Zhu et al., 2025), and can be co-trained on action-free videos, but they incur pixel-level generation cost at inference time unless special acceleration strategies are applied. (C) *WM-as-intermediate reference*: the world model is placed outside the policy loop and produces auxiliary guidance rather than direct action outputs. Dreamitate (Liang et al., 2024), for example, extracts tool trajectories from generated videos and converts them into actions through heuristic planners. GE-Act cross-attends to multi-scale, multi-view latents from GE-Base without re-compression and uses an asynchronous slow–fast schedule for low-latency, precise manipulation.

## 7 CONCLUSION

In this paper, we proposed Genie Envisioner (GE) that unifies egocentric visual representation learning and policy learning within a single closed-loop video generative world model. Through extensive experiments, we demonstrated that GE-Base provides strong spatiotemporal priors via large-scale embodied video pretraining, while GE-Act enables efficient and accurate action policy prediction through asynchronous inference. GE supports both short- and long-horizon tasks, and generalizes across embodiments only with one-hour demonstrations. Together, these components establish a scalable framework that advances the integration of high-level world modeling and low-level control, and lay the groundwork for future video-based neural simulators in robotics.

## 8 ACKNOWLEDGMENTS

This work was partly supported by the National Natural Science Foundation of China under Grant No. 62320106007, and NUS Start-up Grant A-0010106-00-00. We appreciate the AgiBot Genie Team for their invaluable contributions to data collection, real-world evaluation, and the provision of both robotic hardware and software support throughout this research.

## 9 ETHICS AND REPRODUCIBILITY STATEMENT

### 9.1 ETHICS STATEMENT

We follow the ICLR Code of Ethics. **Subjects, data, and safety.** Teleoperation and policy evaluation were done by trained adult researchers. They were paid above-average local rates. We did not collect

PII or biometric data. Any released media will be checked to remove identifying information. We use three data sources: AgiBot-World-Beta under its license, other open academic benchmarks under their licenses, and our own teleoperation data. All robot experiments followed lab safety rules (emergency stop, speed/force limits, safe workspaces, and human–robot separation). We avoid hazardous tasks and add safeguards to reduce risks.

**Fairness, environment, and disclosure.** Our system does not use sensitive personal attributes, but scene or object bias can still appear. We try to reduce this by using diverse tasks and testing across embodiments. We will release checkpoints to support reuse. We will disclose funding and affiliations in the camera-ready. Sponsors did not influence our study. Upon acceptance, we have released code, checkpoints, data schemas, and filtered datasets with licenses and clear docs to help reproduction, together with known limitations, in line with the ICLR Code of Ethics.

## 9.2 Reproducibility Statement

We aim to maximize reproducibility. The main paper specifies model architectures(Section 2.1 and Section 3.1), training pipelines and hyperparameters (Figure 4, Section 2.2, and Section 3.2), and inference strategies (Section 3.3). The appendix details tasks, setups, and qualitative visualizations. We provide an anonymized codebase in the supplementary materials, including config files, data loaders, and evaluation scripts. For datasets, we use open-access academic benchmarks as cited, and we will release our real-world datasets upon acceptance with data schemas, preprocessing scripts, and licenses. Additional qualitative demos are included as supplemental videos.

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

# A APPENDIX

## A.1 TASKS DESCRIPTION

The real-world manipulation tasks include: (1) *Make a sandwich*: sequentially assembling bread, bacon, lettuce, and bread, which tests multi-object coordination, spatial reasoning, and procedural task execution; (2) *Pour a cup of tea*: involving grasping, precise pouring, and repositioning a teapot, highlighting the need for fine-grained motion control and dexterity in fluid manipulation; (3) *Clean the table*: requiring the robot to grasp a wiper and perform consistent wiping motions to remove surface stains, evaluating trajectory stability and compliant force application; (4) *Heat food in the microwave*: operating a microwave door, inserting a bowl, and interacting with buttons, challenging the system's ability to handle articulated objects and multi-stage interface operations; (5) *Pack laundry detergent*: grasping moving detergent bags from a conveyor belt and placing them into boxes, designed to assess dynamic perception, motion tracking, and industrial-scale manipulation.

## A.2 EFFECT OF MULTI-SCALE VISUAL FEATURES CONDITION MECHANISM ON ACTION POLICY

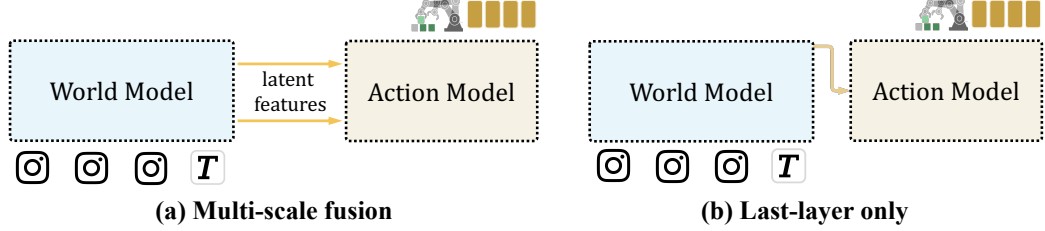

(a) Multi-scale fusion        (b) Last-layer only

Figure 10: **Comparison of GE-Act Performance under Two Variants Visual Condition Strategies.**

Table 2: Effect of visual conditioning on GE-Act. Multi-scale fusion outperforms conditioning on only the last video layer feature.

| Variant | SR ↑ | E2E ↑ |
|---|---|---|
| Last-layer only | 0.64 | 0.81 |
| Multi-scale fusion (ours) | **0.76** | **0.89** |

We study how the choice of visual conditioning affects GE-Act. Prior VLA work commonly conditions the policy on only the last visual layer. In contrast, our design injects multi-scale features aligned across blocks, enabled by the matched DiT structure between GE-Base and the action head. To validate the effectiveness, we compare these two variants as shown in Figure 10, under the same data, training schedule, and inference setup. We validate this setting with the task, *grasping a red cylinder from the table and placing it into a paper cup with fixed position*.

*"Pick up the milk from the refrigerator"*

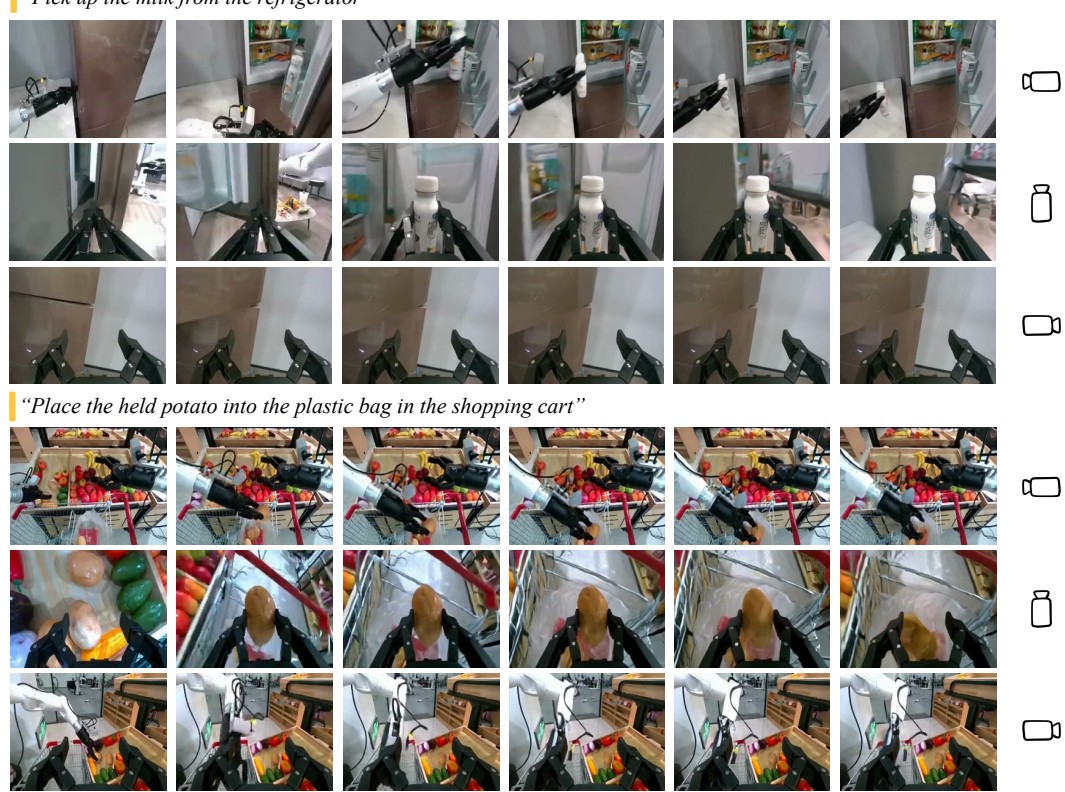

*"Place the held potato into the plastic bag in the shopping cart"*

Figure 11: **Multi-View Robotic Manipulation Videos Generated on AgiBot G1 by GE-Base.** We visualize robotic manipulation sequences generated by GE-Base across two tasks involving varied objects and environments. For each example, videos from three views are presented, *i.e.*, the head-mounted, left-, and right-arm cameras, respectively.

- Last-layer only: action blocks cross-attend to the final video layer.
- Ours : action blocks attend to a scale-aware fusion of video expert latent features aligned across DiT depth.

As shown in Table 2, multi-scale fusion supplies richer cues in a block-aligned manner, yielding more stable grasps and fewer recoverable failures.

### A.3 REAL-WORLD GENERATION VISUALIZATION ON AGIBOT G1

As shown in Figure 11, GE-Base generates multi-view videos that accurately reflect diverse language instructions. The results highlight the model's ability to maintain spatial consistency across views, preserve background and scene structure, and produce stable, step-by-step execution aligned with the instruction semantics.

### A.4 REAL-WORLD MANIPULATION QUALITATIVE RESULTS ON AGIBOT G1

Figure 12 demonstrates GE-Act's ability to execute complex manipulation tasks precisely and reliably from natural language instructions on AgiBot G1.

### A.5 REAL-WORLD GENERATION VISUALIZATION ON AGILEX COBOT MAGIC

As illustrated in Figure 13, our adapted GE-Act model generates coherent, instruction-conditioned multi-view videos for the cloth folding and box folding tasks. These videos accurately capture

*"Use the sponge held in the right arm to wipe the stains clean"*

*"Pour water into the cup on the table"*

*"Make a sandwich"*

*"Packing washing detergent from the conveyor belt"*

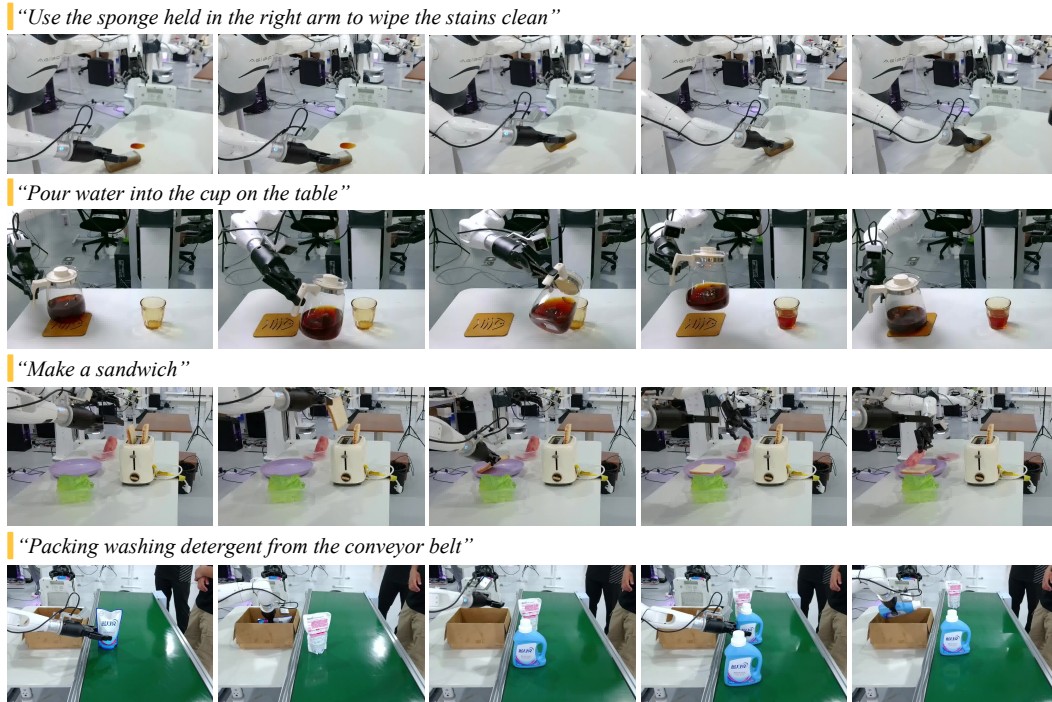

Figure 12: **Visualization of Real-World Robotic Manipulation on AgiBot G1 via GE-Act.** Conditioned on natural language instructions, GE-Act generates and executes action policies on the AgiBot G1 platform. The visual samples demonstrate the model's capability to produce consistent, reliable, and contextually appropriate manipulation behaviors, showcasing its robustness and effectiveness in real-world environments.

both rigid and non-rigid object dynamics with high fidelity. The results demonstrate strong consistency across different camera views and showcase GE-Act's robust handling of complex object deformations.

*"Folding the grey clothes"*

*"Folding the blue clothes"*

*"Folding the box"*

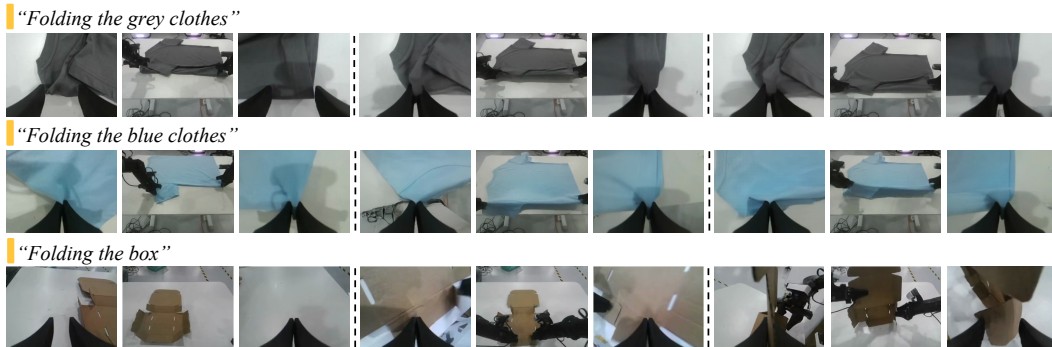

Figure 13: **Multi-View Video Generation on the Agilex Cobot Magic Robotic Platform by GE-Base.** Visualization of instruction-conditioned video generated by GE-Base for two complex folding tasks on the cross-embodiment Agilex Cobot Magic robot. Each row displays temporally sampled frames from a multi-view sequence.

## A.6 REAL-WORLD GENERATION AND MANIPULATION VISUALIZATION ON DUAL FRANKA

Figure 14 illustrates the cloth folding task on the Dual Franka platform, including both the future-space video predictions by GE-Base and the real-world manipulation results executed by GE-Act. The results indicate that GE effectively models task-relevant visual dynamics and generalizes to new embodiments for precise manipulation.

*"Fold the yellow clothes"*

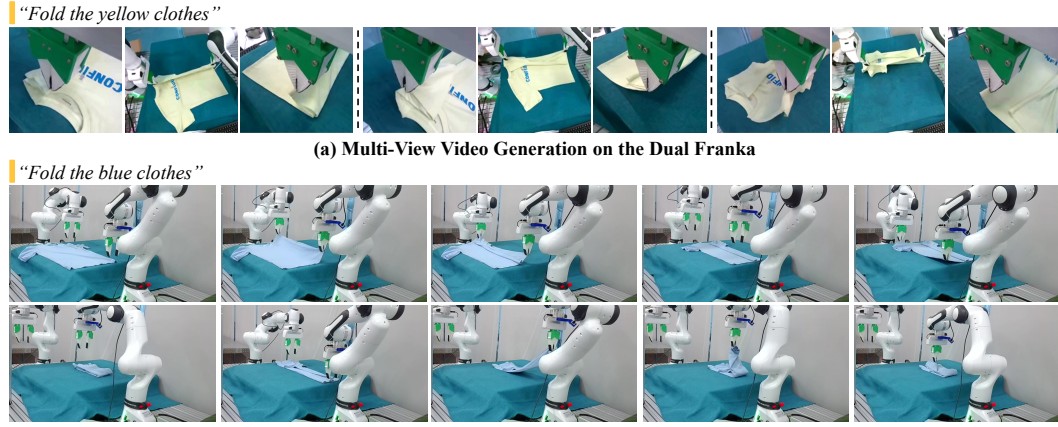

(a) Multi-View Video Generation on the Dual Franka

*"Fold the blue clothes"*

(b) Real-World Robotic Manipulation on the Dual Franka

Figure 14: **Visualization of Robotic Video Generation and Real-World Manipulation on Dual Franka via GE.**

## A.7 LIMITATIONS

This work investigates unified world models for real-world robotic manipulation, coupling embodied video generation (GE-Base) with an action model (GE-Act). While the framework advances spatiotemporal representation learning and low-latency control, several limitations remain:

- Data coverage and source diversity. Pretraining relies primarily on AgiBot-World-Beta (real, dual-arm, multi-view) without incorporating heterogeneous sources such as large-scale web video, broader robot platforms, or high-variance simulation. This limits exposure to diverse embodiments, sensors, and long-tail scene configurations. Although few-shot adaptation shows promise on Dual Franka, Agilex Cobot Magic, and RoboTwin, systematic robustness to OOD embodiments remains underexplored. Future work can integrate multi-source pretraining and study domain-mixing strategies with explicit data provenance controls.

- Embodiment scope and dexterity. Our evaluations focus on upper-body tabletop manipulation with parallel-jaw grippers and do not cover dexterous in-hand manipulation, tool use with tight tolerances, or mobile whole-body behaviors (navigation + manipulation). Extending GE to articulated hands and mobile bases will require richer actuation spaces, contact-aware objectives, and safety-constrained control.

- Real-time video modeling vs. control bandwidth. GE's asynchronous inference reduces latency by running video prediction at 5 Hz and actions at 30 Hz, but it still depends on periodic video denoising and cached latents. Highly dynamic tasks that demand faster updates may require further optimization.

- Compute efficiency and accessibility. GE-Base and GE-Act training uses many GPUs. Although inference is efficient on a single commodity GPU, the pretraining budget may limit adoption. Future work will pursue token pruning, quantization, low-rank adapters, and student models distilled from GE-Base to democratize training and deployment.

  These limitations suggest concrete next steps. And we are leaving these for our future research topics.

### A.8 Usage of GPT in Paper Writing

In the process of writing the paper, we used GPT only for minor refinement during the editing stage. Specifically, we provided the initial draft and used GPT to improve the text by instructing it to "ensure the text is free of grammatical errors" and "refine the logical expression," among other instructions. This approach allowed GPT to enhance the clarity and coherence of the writing while maintaining the original ideas and structure, ensuring the paper met academic writing standards.

