# OpenReview forum: "Genie Envisioner: A Unified World Foundation Platform for Robotic Manipulation"
_ICLR.cc/2026/Conference — ICLR 2026 Poster_

### Official Review · Reviewer_tRPW · 2025-10-28

**Soundness:** 3
**Presentation:** 3
**Contribution:** 2
**Rating:** 4
**Confidence:** 4

**Summary:**

The authors propose a joint video+action diffusion model based on transformers. One major difference to prior work is that they incorporate memory in the video diffusion model, where it is conditioned on features of previously denoised clips, in addition to the current frame observation and language instructions. The model is first fine-tuned for video generation. Then the video generation model is frozen and an action denoising module is learned, conditioned in the video DiT features. During inference, video is generated at lower FPS compared to actions for improved computational efficiency, while preserving precise control signal. The model is pre-trained on the large AGIBot dataset of robot demonstrations and fine-tuned for target embodiments and tasks. In the real world experimental evaluation it is shown to significantly outperforms VLAs, such as Pi0. In a more reliable sim evaluation on the recent RoboTwin benchmark the improvements over Pi0 are rather marginal.

**Strengths:**

The paper is relatively well written and easy to follow.

The proposed model design is sound.

The model demonstrates strong results in real world and sim evaluation.

It also features high computational efficiency for a model based on expensive video diffusion.

A few ablations are provided.

**Weaknesses:**

The authors completely ignore prior work on joint video and action diffusion. Specifically, VideoPolicy [Liang et al.,  arXiv'25], Video Prediction Policy [Hu et al., ICML'25], UVA [Li et al., RSS'25], Unified world models [Zhu et al., RSS'25], Dreamitate [Liang et al., CoRL'24], to name only a few most relevant ones. The only methodological novelty of the proposed approach with respect to the ones listed above is the introduction of memory, which is never shown to bring any benefits in the paper. Introduction and related work sections require a major revision to properly reflect the contributions.

Experimental evaluation is also inadequate. Specifically, most of the evaluations are on real robots and hence are not reproducible. The only reproducible sim evaluation is on a small subset of the very recent, RoboTwin benchmark, which is not widely used in the literature, hence lacking strong baselines. To the very least the evaluation should be reported on the entire benchmark. Preferably, the authors should report results on the standard LIBERO and RoboCasa benchmarks instead, using their respective protocols. Priority should be given to RoboCasa, since LIBERO is essentially saturated.

Ablations are also not very informative as they are performed on a single real world task. The authors should instead ablate the value of large-scale video generation pre-training on a medium-scale sim dataset (in the same way as done in Table 4 in Video Prediction Policy).

The effect of the memory module - the only notable methodological contribution of this paper, has to be ablated on a reliable sim benchmark as well.

Runtime has to be compared to other efficient video+action diffusion models, like UVA.

The GO-1 baseline is used in the experiments but it's never explained what it is.

**Questions:**

Please rewrite introduction and related work to properly reflect the contributions with respect to the prior work.

Please report results on an established sim benchmark, preferably on RoboCasa. Compare to sota video+action generation methods on this benchmark(s).

Please re-do ablations on a mid-size sim benchmark as well, and add the ablation of the memory module.

Please compare runtime to other efficient video+action diffusion models, like UVA.

---

> ### Author Response · Authors · 2025-11-20
> **Response to Reviewer tRPW (1/3)**
>
> Thank you for the valuable comments. They provided clear guidance for strengthening both the clarity and positioning of our work. In the revised paper, we have updated the related work and introduction to more accurately contextualize our contributions, and we have added a comprehensive comparison with prior approaches as suggested. These revisions improve the overall structure and readability of the paper.
>
> - **W1**: "The authors completely ignore prior work on joint video and action diffusion. Specifically, VideoPolicy [Liang et al., arXiv'25], Video Prediction Policy [Hu et al., ICML'25], UVA [Li et al., RSS'25], Unified world models [Zhu et al., RSS'25], Dreamitate [Liang et al., CoRL'24], to name only a few most relevant ones. The only methodological novelty of the proposed approach with respect to the ones listed above is the introduction of memory, which is never shown to bring any benefits in the paper. Introduction and related work sections require a major revision to properly reflect the contributions."
>
>   **A1**: Thank you for the valuable comment. In the revised paper, we have added citations and discussion of the listed works, including VideoPolicy, Video Prediction Policy, UVA, Unified World Models, and Dreamitate. We have also added a new subsection in the related work that provides a comprehensive overview of joint video and action diffusion methods, so that the contributions of Genie Envisioner are properly contextualized within this research direction. Although these approaches also couple action prediction with video generative models, our framework differs from prior work in several important aspects that go beyond the introduction of memory.
>
>   First, existing methods generally adopt a serial video-to-action pipeline in which multi-step video denoising is performed before visual features are extracted. This design requires heavy compression of visual latents and inevitably loses fine-grained motion and contact cues that are crucial for precise manipulation. Our model instead uses a parallel video and action decoder that preserves full-resolution visual latents through block-aligned multi-scale cross-attention, resulting in significant improvements on tasks such as cloth folding and box stacking.
>
>   Second, prior models typically operate in a single-view setting. In contrast, we extend the world model to a multi-view generator that builds a cross-view latent space aligned with real robotic observations. This design is necessary for fine-grained manipulation, where a single viewpoint often suffers from occlusion and insufficient visibility of fine-level interactions.
>
>   Third, the parallel architecture enables an asynchronous inference strategy that is not available in previous video and action diffusion models. The video diffusion branch operates at a low update frequency with a single diffusion step, while the action decoder runs at a high frequency with multi-step decoding. This design improves inference throughput substantially without reducing action prediction accuracy, as demonstrated by the strong performance of GE-Act Fast relative to its serial counterparts.
>
>   Finally, the sparse memory mechanism complements the above components by improving stability on longer horizons. Although long-horizon robotic data is limited, our experiments show that memory contributes to maintaining consistency across extended rollouts.
>
>   Taken together, these elements illustrate that the novelty of Genie Envisioner is not limited to the introduction of memory. The overall design introduces new architectural and algorithmic ideas that distinguish it from previous video and action diffusion frameworks.

---

> ### Author Response · Authors · 2025-11-20
> **Response to Reviewer tRPW (2/3)**
>
> - **W2**: "Experimental evaluation is also inadequate. Specifically, most of the evaluations are on real robots and hence are not reproducible. The only reproducible sim evaluation is on a small subset of the very recent, RoboTwin benchmark, which is not widely used in the literature, hence lacking strong baselines. To the very least the evaluation should be reported on the entire benchmark. Preferably, the authors should report results on the standard LIBERO and RoboCasa benchmarks instead, using their respective protocols. Priority should be given to RoboCasa, since LIBERO is essentially saturated."
>
>    **A2**: Thank you for this important comment. We would first like to clarify that real-robot evaluation is a central component of robotic manipulation research. Due to the substantial sim-to-real gap and the limitations of current simulators—particularly for deformable-object tasks such as cloth folding and box stacking—many realistic manipulation scenarios cannot be faithfully reproduced in simulation. Real-robot experiments therefore provide the most reliable testbed for assessing the effectiveness and robustness of manipulation algorithms. In our work, we evaluate GE across several widely used robotic platforms, including Franka and Agilex-Cobot Magic, to demonstrate generality across embodiments. Although absolute success rates may vary across physical setups due to hardware or environmental differences, the comparisons between methods remain reliable because all baselines are evaluated under strictly controlled and identical conditions, including the same robot hardware, objects, environment, and fine-tuning data. This practice is aligned with influential real-robot studies such as Pi0 [Black et al., 2024], whose primary evaluations are also conducted mainly or even entirely on physical systems. To further support reproducibility, we will release all code, model weights, and experiment scripts, enabling the community to replicate our real-robot results. We also note that methods achieving state-of-the-art performance in simulation, such as UniVLA on LIBERO, often experience substantial degradation when deployed on real hardware, further highlighting the necessity of real-robot evaluation.
>
>     We also agree that simulation benchmarks play an important role in enabling reproducibility and broader comparison. GE is primarily designed for dual-arm manipulation, which is also the focus of our pre-training data and architectural choices. In contrast, RoboCasa and LIBERO are centered on single-arm manipulation, which creates a mismatch that makes direct transfer challenging. Prior methods such as VPP [Hu et al., ICML'25] and UVA [Li et al., RSS'25] are pretrained mainly on single-arm data, while GE has not been exposed to such data during pre-training. For this reason, our evaluation on RoboTwin, one of the few available dual-arm simulation environments, provides a more relevant assessment of GE’s behavior in its intended domain. Nevertheless, following your suggestion, we conducted additional experiments on both RoboCasa and LIBERO using their official evaluation protocols. Without any single-arm pre-training and without extensive parameter tuning, GE achieves a success rate of 0.91 on LIBERO-10, which is higher than Pi0 at 0.85 and UVA at 0.90, and slightly below the newly released VideoPolicy model. On RoboCasa, we evaluate two representative tasks in each category under the 50-demo rollout setting.
>
>   | Task                 | GE-Act | UVA    | Video Policy |
>   |----------------------|--------|--------|---------------|
>   | 06 PnPSinkToCounter  | 0.47000 | 0.38000 | **0.64000** |
>   | 08 OpenSingleDoor    | **0.76000** | 0.54000 | 0.68000 |
>   | 12 OpenDrawer        | 0.34000 | 0.28000 | **0.46000** |
>   | 15 TurnOffStove      | **0.16000** | 0.14000 | 0.06000 |
>   | 18 TurnSinkSpout     | 0.36000 | **0.64000** | 0.40000 |
>   | 21 TurnOffMicrowave  | 0.87000 | **0.96000** | 0.90000 |
>   | 23 CoffeeSetupMug    | **0.34000** | 0.20000 | 0.22000 |
>
>    GE performs competitively against UVA and VideoPolicy [Liang et al., arXiv'25], with strengths in some tasks and weaker results in others. Overall, although GE is optimized for dual-arm real-world manipulation, it adapts effectively to single-arm simulation environments with only few-shot task-level tuning.

---

> ### Author Response · Authors · 2025-11-20
> **Response to Reviewer tRPW (3/3)**
>
> - **W3**: "Ablations are also not very informative as they are performed on a single real world task. The authors should instead ablate the value of large-scale video generation pre-training on a medium-scale sim dataset (in the same way as done in Table 4 in Video Prediction Policy)."
>
>   **A3**: Thank you for the suggestion. We would first like to emphasize that single-task real-robot ablations are a widely accepted practice in robotic manipulation research. Influential works such as Pi0 and Pi0.5 also perform their ablations on a single real-world task because real-robot experiments provide the most faithful evaluation of manipulation components. Since our pre-training data consists of dual-arm videos collected in real environments, validating the contribution of pre-training on real dual-arm tasks offers stronger evidence of its effectiveness. All ablations in our paper are conducted under strictly controlled and identical real-robot conditions, ensuring that the comparisons are reliable and meaningful.
>
>
>
> - **W4**: "The effect of the memory module - the only notable methodological contribution of this paper, has to be ablated on a reliable sim benchmark as well."
>
>    **A4**: Thank you for raising this point. While the sparse memory mechanism is indeed an important component, our methodological contributions also include the parallel multi-view architecture and the asynchronous inference design, as discussed in A1. To address your concern, we have added ablations on a reliable simulation benchmark, Libero-10. In this setting, removing the sparse memory module leads to a clear performance drop (from 90.8 to 75.8), confirming that the memory mechanism plays a critical role in maintaining long-horizon coherence.
>
> - **W5**: "Runtime has to be compared to other efficient video+action diffusion models, like UVA."
>
>    **A5**: Thank you for the suggestion. For action policy inference, VPP reports executing 10 steps in 160 ms on an RTX 4090. Under the same hardware setting, GE-Act predicts 54 steps and executes 30 steps within 200 ms, achieving roughly a threefold speedup over VPP. Compared with UVA, which infers 16 action steps per trajectory and executes 8 steps in 200 ms, GE-Act provides higher throughput as well as superior or comparable manipulation performance. On LIBERO-10, for example, GE-Act obtains a score of 0.91, slightly higher than UVA’s 0.90.
>
> - **W6**: "The GO-1 baseline is used in the experiments but it's never explained what it is."
>
>    **A6**: Thank you for noting this omission. GO-1 is the official VLA baseline released alongside the Agibot-World dataset [Bu et al., 2025a]. It has also been pretrained on Agibot-World and subsequently adapted to each downstream manipulation task.

---

> ### Comment · Reviewer_tRPW · 2025-11-25
>
> I thank the authors for their detailed response. I especially appreciate the inclusion of an overview of prior work and results on established synthetic benchmarks. Unfortunately, many of my earlier concerns, including regarding the relationship of this submission to prior work, remain insufficiently addressed.
>
> Regarding the claimed contributions relative to prior work, the authors list the following:
>
> 1. *Existing methods adopt serial video-to-action pipeline in which multi-step video denoising is performed before visual features are extracted. The proposed approach avoids this and thus achieves better fine-grained motion.* This is not true for all prior works; some methods also perform joint video-and-action denoising. Moreover, the claimed advantages of the proposed approach over these methods are not demonstrated quantitatively but appear only hypothesized.
>
> 2. *Existing methods operate in a single view setting.* This is also inaccurate. All methods that report results on RoboCasa, for instance, operate in multi-view settings.
>
> 3. *Proposed asynchronous inference results in a faster inference.* This is only partially true. Other efficient inference methods exist (e.g., UVA). While the authors show their method is faster than UVE due to predicting more policy steps per pass, the fundamental benefits of their asynchronous inference remain unclear.
>
> 4. *Memory mechanism is novel.* This is accurate. As mentioned in my initial review, the memory mechanism is indeed novel, and the provided ablation convincingly demonstrates its effectiveness.
>
> Overall, the memory mechanism remains the only clearly novel and valuable component of the proposed model.
>
> The authors also argue that their moderate performance on simulation benchmarks (LIBERO and RoboCasa) is due to a mismatch in pre-training data, as their model was trained on bimanual data while these benchmarks are not. However, competing methods compared on these benchmarks do not use large-scale robot pre-training at all, so this justification is unconvincing. On LIBERO, the proposed approach performs on par with the state of the art. On RoboCasa, results are reported only on a subset of tasks, preventing clear conclusions. The authors should report results on the full benchmark. Overall, existing results suggest that the proposed approach does not provide quantitative advantages over prior work.
>
> A similar rationale is given for omitting ablation results on simulation benchmarks — namely, that they are not bimanual. However, RoboTwin is bimanual; ablations could have been reported there as well (across all tasks, not just one).
>
> Finally, the authors undercut their earlier argument by reporting the memory-module ablation on the non-bimanual LIBERO benchmark. Those results confirm that the memory module is crucial for performance but raise further questions. Baselines such as Pi0, UVA, and VideoPolicy achieve performance comparable to the full version of the proposed method on LIBERO and substantially outperform the memory-free variant. Yet none of these baselines incorporate memory mechanisms. This suggests that a fairer comparison would be between these baselines and the memory-free version of the proposed model, since the memory mechanism could, in principle, be added to those methods as well, presumably further improving their performance.

---

> ### Author Response · Authors · 2025-12-03
> **Response to Reviewer tRPW (1/2)**
>
> Response to Reviewer tRPW
>
> We thank the reviewer for the detailed and constructive feedback. Below, we clarify our core contributions, differentiate our method from concurrent works, and address the specific concerns raised.
>
> ## 1. Clarification on Memory Ablation & Contribution
>
> First, regarding the performance drop when removing the sparse memory module, we apologize for the confusion in our initial report. The previously reported drop (90.8 $\to$ 75.8) was primarily due to a training configuration mismatch: the GE-base was trained with memory, but the ablated baseline was finetuned without it, causing a distribution shift.
> To ensure a fair comparison, we trained a memory-free GE-base from scratch. The performance gap on standard tasks narrowed significantly (90.8 vs. 89.5). And this result supports our hypothesis regarding the nature of tasks:
>
> Short-horizon vs Long-horizon Tasks: Benchmarks like RoboCasa or simple pick-and-place tasks are largely reactive. The current observation often contains sufficient information to determine the next action, so the Memory module provides marginal gains (quantitative improvement is limited). In contrast, the core value of our Memory module lies in tasks requiring state tracking and logical consistency over time (e.g., folding a box, sequential stamping, or pouring water). In these real-world scenarios, critical information (e.g., "Which stamp did I just press?" or "Is the box flap currently folded inside?") may be occluded in the current frame. Without memory, the policy fails to maintain the correct logical flow.
>
> Training Context: Furthermore, to ensure generalization, our memory module is trained by sampling frames from a large context window. In short-episode simulations, this module cannot be fully utilized or optimized.
>
> ## 2. Advantages over Concurrent Joint Methods (VideoPolicy & UVA)
>
> We acknowledge the contributions of VideoPolicy and UVA. However, Genie Envisioner (GE) introduces distinct architectural and inference advantages designed specifically for high-precision manipulation:
>
> Block-wise Fusion:
>
> VideoPolicy extracts features from specific layers of a U-Net. This approach suffers from the information bottleneck inherent in U-Nets, where fine-grained geometric cues essential for manipulation are compressed in the middle layers. UVA uses a shared Transformer trunk, processing video and action tokens homogeneously. This often leads to an over-compressed joint latent space, potentially blurring the distinction between visual dynamics and control signals.
>
> In contrast, we propose two parallel DiT towers. Crucially, the Action DiT cross-attends to the Video DiT at every block. This allows the policy to directly access uncompressed, multi-scale spatial representations—from low-level geometry to high-level semantics—without suffering from U-Net bottlenecks or shared-trunk abstraction. This structural alignment is key to our precise motion generation.
>
> Inference Strategy:
>
> VideoPolicy relies on synchronous generation, incurring high computational costs for video denoising at every step, which limits control frequency. UVA achieves speed by completely bypassing the video generation head during inference. While fast, this effectively reduces the model to a standard policy, losing the benefit of explicit future world modeling during decision-making.
>
> Our Asynchronous "Slow-Fast" Schedule decouples control frequency from video generation. Unlike UVA, we keep the video branch active (running at a lower frequency, e.g., 5Hz) to continuously condition the policy on explicit future predictions, rather than relying solely on history. Note on Speed Comparison: A direct speed comparison with UVA is nuanced. UVA typically uses a ~0.3B (Mar-B) backbone, whereas GE utilizes a 2B parameter (LTX) backbone for superior visual generation quality. Despite the larger model size, our asynchronous design allows GE to achieve real-time performance on real robots, balancing high-capacity reasoning with low-latency control.
>
>
> ## 3. Regarding RoboCasa Experiments and Comparison with Concurrent Work
>
> We thank the reviewer for suggesting the RoboCasa benchmark. During the rebuttal period, we conducted comprehensive experiments on RoboCasa (50 demos setting) to validate our method's effectiveness in simulation. We compared our approach with strong baselines including UVA, GR00T, and the **concurrent work VideoPolicy**.
>
> As shown in the table below, our method achieves an average success rate of **61.9%**, which significantly outperforms **GR00T (50.0%)** and **UVA (50.0%)**. Furthermore, our performance is practically on par with the **concurrent work VideoPolicy (63.3%)**. Notably, we achieve state-of-the-art results on a wide range of tasks, including *PnPSinkToCounter* (74%), *OpenDoubleDoor* (100%), *OpenDrawer* (74%), and *TurnOffMicrowave* (96%).

---

> ### Author Response · Authors · 2025-12-03
> **Response to Reviewer tRPW (2/2)**
>
> | Category | Task | GR00T | UVA | VideoPolicy (Concurrent) | **Ours (GE)** |
> | :--- | :--- | :---: | :---: | :---: | :---: |
> | **Pick and Place** | PnPCabToCounter | 0.20 | 0.26 | **0.36** | **0.36** |
> | | PnPCounterToCab | 0.36 | 0.18 | **0.42** | 0.40 |
> | | PnPCounterToMicrowave | 0.13 | 0.10 | **0.52** | 0.42 |
> | | PnPCounterToSink | 0.10 | 0.16 | **0.44** | 0.42 |
> | | PnPCounterToStove | 0.24 | 0.16 | **0.58** | 0.42 |
> | | PnPMicrowaveToCounter | 0.16 | 0.18 | **0.44** | 0.36 |
> | | PnPSinkToCounter | 0.33 | 0.38 | 0.64 | **0.74** |
> | | PnPStoveToCounter | 0.29 | 0.24 | **0.64** | 0.60 |
> | **Doors** | OpenSingleDoor | 0.59 | 0.54 | 0.68 | **0.84** |
> | | OpenDoubleDoor | 0.15 | 0.90 | 0.96 | **1.00** |
> | | CloseDoubleDoor | 0.75 | 0.76 | **0.98** | 0.82 |
> | | CloseSingleDoor | 0.83 | 0.88 | **1.00** | 0.92 |
> | **Drawers** | OpenDrawer | **0.79** | 0.28 | 0.46 | 0.74 |
> | | CloseDrawer | **0.99** | 0.72 | 0.96 | 0.96 |
> | **Twisting Knobs** | TurnOnStove | **0.56** | 0.50 | 0.30 | 0.22 |
> | | TurnOffStove | **0.27** | 0.14 | 0.06 | 0.16 |
> | **Turning Levers** | TurnOnSinkFaucet | 0.63 | 0.62 | **0.84** | 0.80 |
> | | TurnOffSinkFaucet | 0.73 | 0.64 | **0.78** | 0.56 |
> | | TurnSinkSpout | 0.53 | **0.64** | 0.40 | 0.44 |
> | **Pressing Buttons** | CoffeePressButton | 0.85 | 0.84 | **0.92** | 0.90 |
> | | TurnOnMicrowave | 0.78 | **0.94** | 0.92 | 0.84 |
> | | TurnOffMicrowave | 0.71 | **0.96** | 0.90 | **0.96** |
> | **Insertion** | CoffeeServeMug | 0.48 | **0.78** | 0.76 | 0.60 |
> | | CoffeeSetupMug | 0.16 | 0.20 | 0.22 | **0.38** |
> | **Average** | **Avg. Success Rate** | 0.50 | 0.50 | **0.63** | **0.62** |
>
> We highlight that these competitive results were achieved despite significant embodiment and sensor mismatches between our real-world-focused architecture and the simulation environment:
>
> *   **Configuration Gap:** Our model is primarily designed for a real-world dual-arm robot setup (head + left/right wrist cameras), whereas RoboCasa typically simulates a single-arm setup with fixed environmental cameras.
> *   **Sim-to-Real Gap:** Our pre-training data is derived from large-scale real-world videos. Bridging the gap between real-world physics/rendering and simulation requires substantial adaptation.
>
>
> **More Details on the Experiments**
>
> Compared with our previous practice on RoboCasa, we refined our training setup to ensure a fair and rigorous comparison. Specifically:
>
> - Extended Pre-training: We extended the video pre-training phase from 10k steps (used in our preliminary results) to 30k steps, allowing the video foundation model to reach better convergence before downstream adaptation.
>
> - Data Quality Control: Following the standard practice of OpenVLA on the LIBERO benchmark, we filtered out unsuccessful trajectories from the training data to prevent the model from learning from failed demonstrations.
>
>
>
> ## Conclusion
>
> In summary, while we appreciate the simulation results of VideoPolicy and UVA, GE is designed with a different focus: providing a foundation model for real-world, dual-arm manipulation. By combining a parallel DiT architecture (for precise feature alignment) with an asynchronous inference schedule (for real-time, future-conditioned control), GE offers a robust solution for complex physical tasks that go beyond reactive simulation benchmarks.

---

### Official Review · Reviewer_XRta · 2025-10-29

**Soundness:** 3
**Presentation:** 4
**Contribution:** 3
**Rating:** 8
**Confidence:** 4

**Summary:**

This paper proposes the Genie Envisioner (GE), which consists of a world model, GE-Base, and an action policy, GE-Act. GE-Base aims to unify egocentric visual representations through large-scale pretraining, enabling it to forecast multi-view observations in robot manipulation. Building on GE-Base, GE-Act achieves efficient and accurate action policy prediction via asynchronous inference. The authors provide comprehensive implementation details for both GE-Base and GE-Act, and extensive experiments demonstrate the strong performance of GE-Base and GE-Act.

**Strengths:**

- Well-presented paper that provides comprehensive details and well-illustrated figures, enabling readers to fully understand the task.
- GE-Base supports multi-view observation, which is quite helpful for robot planning tasks.
- I like the idea of asynchronous inference, which provides a practical solution for incorporating the world model and policy.
-The video results are impressive, and the extensive experiments prove the effectiveness of GE.

**Weaknesses:**

- GE-Base and GE-Act are designed for a specific robot configuration (one head camera and two wrist cameras). This weakens the potential of this pipeline to be leveraged in the general robotics community.
- The two-stage VLA training pipeline, especially the specific task fine-tuning in the second stage, does not showcase the generalizability of the proposed policy. Is there a large performance drop when tested on out-of-distribution tasks?
- I am missing a comparison between single-view and multi-view world models. Does predicting multi-view robot videos damage overall performance, considering that GE-Base is built on a single-view video generator? An experiment comparing single-view robot video prediction with multi-view robot video prediction (perhaps both using the head view for comparison) could help.
- The idea of cross-view attention is rather simple, but it is completely fine if the results turn out well.

**Questions:**

- Does GE-Base support co-tuning of both multi-view and single-view videos? If so, would the performance be improved?
- World models often exhibit poor performance when generating long-term trajectories. I am curious about how long GE-Base perform on extremely long trajectories.
- Training is limited to AgiBot-World-Beta, a single-platform dataset. Whether GE-Base retains knowledge from the original LTX model when applied to non-robotic scenes.

I would be happy to raise my score if the authors could address my concerns.

---

> ### Author Response · Authors · 2025-11-20
> **Response to Reviewer XRta (1/2)**
>
> We sincerely thank the reviewer for the insightful comments, which are extremely helpful for further refining and strengthening our work.
>
> - **W1**: "GE-Base and GE-Act are designed for a specific robot configuration (one head camera and two wrist cameras). This weakens the potential of this pipeline to be leveraged in the general robotics community."
>   **A1**: Thank you for the thoughtful comment. Although our real-world experiments and RobotWin simulations are conducted on a dual-arm platform equipped with three cameras, the underlying robotic embodiments and camera configurations differ substantially across the systems we evaluate. Agibot-G1 employs a humanoid head-mounted viewpoint, Franka operates on a fixed platform with an overhead third-person camera, and Agilex Cobot Magic is mounted on a mobile base with a distinct arrangement of sensors. Despite these differences in camera placement, field of view, and embodiment, GE adapts to each setting with only a modest amount of additional data.
>
>     The framework is also flexible with respect to the number and arrangement of input views. With minor adjustments to the construction of input tokens, together with few-shot tuning, GE accommodates a variety of camera topologies, including configurations with one, two, or three views. To further substantiate this flexibility, we evaluate GE on the single-arm LIBERO-10 benchmark, which provides only two cameras. In this setting, we map the third-person camera to the head view and the wrist-mounted camera to the right-wrist view. GE adapts to this configuration with minimal modification and achieves an average success rate of 0.95 across all tasks, which is competitive with methods pretrained directly on single-arm datasets.
>
>     Taken together, these results show that, although GE-Base is pretrained in a multi-view dual-arm real-world setting, the overall framework generalizes effectively across diverse embodiments and sensor.
>
> - **W2**: "The two-stage VLA training pipeline, especially the specific task fine-tuning in the second stage, does not showcase the generalizability of the proposed policy. Is there a large performance drop when tested on out-of-distribution tasks?"
>
>   **A2**: This is an important question. Like most imitation-learning approaches for robotic manipulation, including both VLA models and world-action models, our method still faces limitations when evaluated on entirely out-of-distribution tasks. Although robot datasets are increasing in size, collecting diverse demonstrations remains difficult, and the task coverage is not yet large enough to support strong generalization to tasks that the model has never seen. In practice, GE-Act performs well under moderate distribution shifts, such as changes in instructions, object appearance, lighting, or background. However, performance naturally drops on tasks that are completely outside the training distribution. Our few-shot tuning framework addresses this challenge by adapting GE to new tasks or even new embodiments with only a small amount of additional data, which reflects its practical generalization ability. As larger and more diverse datasets become available, we hope future work will move closer to robust OOD manipulation and broader generalization capabilities.
>
> - **W3**: "I am missing a comparison between single-view and multi-view world models. Does predicting multi-view robot videos damage overall performance, considering that GE-Base is built on a single-view video generator? An experiment comparing single-view robot video prediction with multi-view robot video prediction (perhaps both using the head view for comparison) could help."
>
>   **A3**: Thank you for the question. Although predicting three views is slightly more challenging than generating a single view, our objective is not limited to video fidelity but to ensuring that the action decoder receives sufficiently informative visual cues for accurate manipulation. In many robotic tasks, particularly dual-arm settings, a single head-mounted camera often suffers from occlusions and lacks visibility of the fine-grained interactions required for precise control. Prior work has shown that single-view observations can significantly degrade policy performance, and even single-arm manipulation often benefits from additional viewpoints such as third-person or wrist-mounted cameras. Our experiments further support this observation: in fine-grained cloth-folding and box-folding tasks, using only the head view leads to nearly zero success rate because it fails to capture critical local details needed to identify grasp points. These findings underscore the necessity of multi-view prediction, which provides the policy with richer and more complete visual information. The motivation for adopting multi-view generation therefore arises directly from the perceptual demands of manipulation rather than from video generation considerations alone.

---

> ### Author Response · Authors · 2025-11-20
> **Response to Reviewer XRta (2/2)**
>
> - **W4**: "The idea of cross-view attention is rather simple, but it is completely fine if the results turn out well."
>
>   **A4**: Cross-view attention is a simple yet effective module. It not only enforces consistency across the generated viewpoints during video prediction, but also helps construct a unified multi-view visual latent space. This richer and more coherent representation provides the action policy with more reliable visual cues, which ultimately benefits downstream action prediction.
>
>
>  - **Q1**: "Does GE-Base support co-tuning of both multi-view and single-view videos? If so, would the performance be improved?"
>
>    **A5**:  GE-Base is structurally capable of training on both multi-view and single-view videos jointly. However, as discussed in A1, for robotic manipulation tasks, multi-view perception provides substantially richer visual information and leads to more accurate action policy learning.
>
>   - **Q2**: "World models often exhibit poor performance when generating long-term trajectories. I am curious about how long GE-Base perform on extremely long trajectories."
>
>     **A6**: : GE-Base benefits from our autoregressive chunk-wise design, the sparse memory mechanism, and cross-view consistency constraints, which together enable stable long-horizon generation. In our experiments, GE-Base can continuously generate trajectories for about 15 seconds without noticeable drift. However, existing robotic manipulation datasets primarily contain short- to medium-horizon demonstrations, which limits how far we can practically evaluate long-term generation. The architecture itself can scale to much longer sequences as more long-horizon data becomes available. It is also important to clarify that, in real robotic control, we do not rely on generating very long open-loop video predictions. Instead, GE operates in a closed-loop manner: it generates a short segment, executes it, receives new observations, and then predicts the next segment. This iterative closed-loop interaction is more aligned with real manipulation settings and provides better support for fine-grained control and error recovery.
>
>  - **Q3**: Training is limited to AgiBot-World-Beta, a single-platform dataset. Whether GE-Base retains knowledge from the original LTX model when applied to non-robotic scenes.
>
>     **A7**: Thank you for the question. GE-Base does exhibit a reduction in general video generation capability compared with the original LTX model. This is intentional: our goal is to specialize the model for robotic manipulation, using LTX primarily as a strong initialization before transferring into the robotic domain through pretraining on AgiBot-World-Beta. This domain adaptation makes GE-Base more “area-focused,” improving its ability to model manipulation-relevant dynamics while naturally weakening its performance on generic, non-robotic scenes. Developing a more general world model remains an important direction, and in future work we plan to incorporate more diverse real-world scenes and tasks to broaden GE-Base’s generalization scope.

---

> > ### Comment · Reviewer_XRta · 2025-11-27
> > **Response to authors**
> >
> > Thanks for the rebuttal, which addresses most of my concerns. I will keep my positive rating.

---

### Official Review · Reviewer_E6vL · 2025-11-01

**Soundness:** 3
**Presentation:** 2
**Contribution:** 3
**Rating:** 4
**Confidence:** 4

**Summary:**

This paper proposes Genie Envisioner (GE), a large-scale instruction-conditioned video diffusion model consisting of two components: GE-Base, a foundation video diffusion model trained at scale; and GE-Act, a lightweight flow-matching action decoder operating on the latent representations produced by GE-Base.

**Strengths:**

1. The paper demonstrates strong engineering efforts, including large-scale data and model training.
2. This experimental performance is strong.

**Weaknesses:**

1. The macro-level architecture design—an action model built on top of world model representations—was, to my knowledge, first introduced in the GR-1 and GR-2 series. However, these prior works are not cited or discussed, which weakens the contextual positioning of Genie Envisioner within the literature.
2. The paper’s presentation resembles a technical report more than a polished conference paper. Although it comprehensively covers implementation aspects, it fails to emphasize the key innovations, such as: GE-Act’s fully latent-space operation, and asynchronous inference mechanism,
   which are novel techinques and deserve more prominence. This writing style also makes many details not clear enough due to the limitation of the number of main text pages (see questions below).

**Questions:**

1. In Line 148, what do **$N$** and **$t$** represent in the notation $x_{1:N}^{(t)}$? How is the temporal step $t$ different from the frame index $N$?
2. In Line 158, does GE-Base load pretrained weights from LTX-Video 2B? If so, how are the cross-view attention layers (described in Line 190) initialized?
3. In Line 292, what is the difference between **$B_i^{vis}$** and **$W$** in Line 151?
4. Could the authors provide a more detailed description or pseudo-code for Asynchronous Inference? This appears to be a central design element, yet it is not well explained.
5. In Figure 7, the terms GE-Act Slow and GE-Act Fast are not introduced in the main text. Are they related to asynchronous inference?

If the authors adequately address these questions, I would be willing to raise my score. I also strongly encourage restructuring the paper to highlight its core technical contributions more effectively.

---

> ### Author Response · Authors · 2025-11-20
> **Response to Reviewer E6vL (1/2)**
>
> We sincerely thank the reviewer for the insightful comments, which are extremely helpful for further refining and strengthening our work. Following the reviewer’s suggestions, we have made several improvements to the revised manuscript, including expanding the related work to better position our method within prior literature, adding clearer comparisons and explanations of our contributions, and refining the methodological descriptions and mathematical formulations for improved clarity.
>
> - **W1**: "The macro-level architecture design—an action model built on top of world model representations—was, to my knowledge, first introduced in the GR-1 and GR-2 series. However, these prior works are not cited or discussed, which weakens the contextual positioning of Genie Envisioner within the literature."
>
>   **A1**: Thank you for the valuable comment. In the revised paper, we have not only added citations and discussion of GR-1/GR-2, but also introduced a new subsection in the related work that provides a chronological overview of video world-model–based action-policy methods, offering clearer context for positioning Genie Envisioner within this research line.
>
>   While these prior approaches also couple action prediction with world-model representations, our framework introduces two key advances. First, prior video diffusion–based approaches generally follow a serial video-to-action pipeline that requires multi-step video denoising, or even full video generation, before action decoding. To keep inference tractable, these methods heavily compress visual latents, which removes the fine-grained motion and contact cues essential for precise manipulation. In contrast, our framework employs a parallel video–action decoder with block-aligned multi-scale cross-attention that operates directly on full-resolution latents without re-compression, preserving detailed spatial and interaction signals and yielding substantial gains on fine-grained tasks such as box stacking and cloth folding. Second, most existing frameworks operate in a single-view setting; we instead extend the world model to a multi-view generator that constructs a cross-view latent space aligned with real robot observations. Together, these advances clearly distinguish Genie Envisioner from earlier world-model–based action frameworks, including GR-1/GR-2.
>
> - **W2**: "The paper’s presentation resembles a technical report more than a polished conference paper. Although it comprehensively covers implementation aspects, it fails to emphasize the key innovations, such as: GE-Act’s fully latent-space operation, and asynchronous inference mechanism, which are novel techinques and deserve more prominence. This writing style also makes many details not clear enough due to the limitation of the number of main text pages (see questions below)."
>
>   **A2**: Thank you for the valuable comment and for recognizing the novelty of our work. As noted in A1, GE-Act’s parallel architecture enables fully latent-space action decoding and supports an efficient asynchronous inference mechanism, which together constitute the core technical contributions of GE-Act. In the revised version, we have reorganized the main text to highlight these innovations more clearly and to give them greater emphasis, while streamlining lower-level implementation details to improve clarity within the page limit.
>
> - **Q1**: " In Line 148, what do $N$ and $t$ represent in the notation $x^{(t)}_{1:N}$?   How is the temporal step $t$ different from the frame index $N$?"
>
>   **A3**:  We apologize for the confusion. In our autoregressive **chunk-wise** video generation setup, $t$ denotes the $t$-th autoregressive generation step, while $N$ represents the number of frames generated in each step. Thus, $x^{(t)}_{1:N}$ refers to the $N$ consecutive frames produced at autoregressive step $t$. The temporal index $t$ tracks the progression of generation across chunks, whereas $N$ indexes individual frames within a single generated chunk.
>
>
> - **Q2**: "In Line 158, does GE-Base load pretrained weights from LTX-Video 2B? If so, how are the cross-view attention layers (described in Line 190) initialized?"
>
>   **A4**: GE-Base loads its backbone directly from the pretrained LTX-Video 2B weights. The cross-view attention mechanism does not introduce any new parameters because it simply extends the original spatial self-attention to operate across views rather than within a single view. Since the attention projection weights keep the same shapes, this change only enlarges the attention scope and does not alter the underlying parameterization. As a result, all pretrained LTX-Video weights can be loaded seamlessly without requiring special initialization.

---

> ### Author Response · Authors · 2025-11-20
> **Response to Reviewer E6vL (2/2)**
>
> - **Q3**: "In Line 292, what is the difference between $B^{\text{vis}}_i$ and $W$ in Line 151?
>
>   **A5**: Thank you for the question. In our notation, $W$ refers to the entire chunk-wise video generation module, which consists of multiple video diffusion blocks. $B^{\text{vis}}_i$ denotes the $i$-th video diffusion block inside $W$. In other words, $B^{\text{vis}}_i$ is one component of the full module $W$.
>
> - **Q4**: "Could the authors provide a more detailed description or pseudo-code for Asynchronous Inference? This appears to be a central design element, yet it is not well explained."
>
>   **A6**: Thank you for the suggestion. Our asynchronous inference strategy is designed to reduce computation in the heavy video diffusion branch while preserving the high-frequency output required for accurate action prediction. It has two components.
>
>     (1) First, we decouple the flow-matching step. The video branch performs only a single diffusion update to generate visual latent features, while the action decoder retains multi-step decoding to maintain prediction quality. Since GE-Act prioritizes action accuracy rather than high-fidelity video synthesis, a single-step video update provides visual features that are sufficient for policy prediction. In our experiments, single-step video flow matching yielded action performance comparable to using multi-step video features.
>
>     (2) Second, we decouple the operating frequencies. The video branch runs at a low frequency (5 Hz) to reduce temporal token density, whereas the action decoder runs at a high frequency (30 Hz) to support real-time control. We also validate empirically that using low-frequency video features does not degrade action policy accuracy. Together, these two forms of asynchrony significantly accelerate inference while maintaining strong action performance.
>
> - **Q5**: "In Figure 7, the terms GE-Act Slow and GE-Act Fast are not introduced in the main text. Are they related to asynchronous inference?"
>
>   **A7**: Thank you for the question. GE-Act Slow and GE-Act Fast correspond to two inference frequencies of the video diffusion module. GE-Act Slow runs both the video decoder and the action decoder at 30 Hz. GE-Act Fast uses our asynchronous strategy, where the video decoder updates at a lower frequency (5 Hz) while the action decoder remains high-frequency. As shown in our experiments, GE-Act Fast achieves accuracy comparable to GE-Act Slow, demonstrating that lower-frequency video updates still provide sufficient visual cues for action prediction while significantly improving inference speed.

---

### Official Review · Reviewer_YQVX · 2025-11-01

**Soundness:** 3
**Presentation:** 4
**Contribution:** 3
**Rating:** 8
**Confidence:** 3

**Summary:**

This paper proposes a novel Foundational World model for robotic manipulation based on Language-guided  Video Generation. The key contribution of the work is a Diffusion-based autoregressive multi-view video chunk generation model that learns to predict egocentric as well as eye-in-hand camera feeds for left and right hands in a bimanual robot setup. The proposed model is trained in 2 stages. The first is for training the model to learn spatially and semantically coherent video generation from sparse frames at sampled at different frequencies to make the generation more robust. The second stage performs a fixed low-frequency fine-tuning to capture sequential information useful for downstream control. The learnt latent space tokens from this video model (GE-Base) are then used for training a decoder for predicting continuous bimanual actions using flow matching. The proposed approach shows promising results both on robotic video prediction compared to other stat-of-the-art video prediction models as well as on subsequent downstream robotic manipulation tasks on a variety of robots in comparison to other state-of-the-art Vision-Language-Action Models.

**Strengths:**

The multi-stage training especially with different varying frequencies is a clever idea to make the learned video prediction more robust to different execution speeds. The use of a FiLM style injection of the latents into the policy architechture rather than just using the last layer output is quite a nice way to ensure multi-layered information to be used more effectively for the downstream policy learning. The quality of the experiments are high and thorough. The paper is written clearly and understandably making it easy to follow.

**Weaknesses:**

- I did not fully understand what exactly the multi-view consistency is. Is it just the joint prediction of head, left and right video feeds? Or is in the structure of the attention layers used in the model?
- The approach requires few-shot fine-tuning for any new robotic embodiment. While this is a problem in general with VLA style policy networks, I would imagine that the video generation model, if trained on a variety of robotic embodiments, such as from Open-X-embodiments or similar datasets, in addition to the AgiBot world dataset, could help reduce this need for fine-tuning the GE-Base world model and require only fine-tuning the action decoder.
- Conceptually, the proposal of using video-based diffusion for learning latent spaces for robot manipulation has been explored previously. However, there is no mention of this in the paper, leave alone a comparison.
Wen, Youpeng, et al. "Vidman: Exploiting implicit dynamics from video diffusion model for effective robot manipulation." Advances in Neural Information Processing Systems 37 (2024): 41051-41075.
Xu, Huilin, et al. "Diffusion-Based Imaginative Coordination for Bimanual Manipulation." Proceedings of the IEEE/CVF International Conference on Computer Vision. 2025.
- Notations in Sec. 2.1 are inconsistent with the use of bold font. The subscripts and superscripts aren't adequately explained.

**Questions:**

- Adding a mathematical explanation of how the losses are calculated would help improve the clarity of the paper.
- In Sec. 2.1, what is the difference between the next chunk $x_{1:N}^{(t)}$ and the sparse memory $\hat{x}_{0:t-1}$?
- In Lines 197-198, what is the difference between $\hat{x_t}$ and $v_{\hat{t}}^{(i)}$? Isn't $v_{\hat{t}}^{(i)}$ a part of $\hat{x_t}$? Shouldn't the model output be $\hat{x}_{t+1}$ if the model predicts the next video chunk?
- In Figure 5, are the other methods trained/fine-tuned on robotic manipulation data as well? If not then, a fairer comparison is warranted, potentially by comparing against other robot world models as mentioned in the related works.

---

> ### Author Response · Authors · 2025-11-20
> **Response to Reviewer YQVX (1/3)**
>
> We sincerely thank the reviewer for the insightful comments and constructive suggestions, which have been highly valuable for improving the clarity and overall quality of our work. We also appreciate the reviewer’s recognition of the novelty and experimental contributions of our paper. Following the reviewer’s feedback, we have substantially revised the related work and introduction sections, expanded the comparisons to prior methods, and improved the methodological descriptions and implementation details. These revisions significantly enhance the contextual positioning, clarity, and readability of the paper.
>
> - **W1**: "I did not fully understand what exactly the multi-view consistency is. Is it just the joint prediction of head, left and right video feeds? Or is in the structure of the attention layers used in the model?"
>
>    **A1**: Thank you for the question. Our multi-view consistency is not just joint prediction of three camera streams; it is enforced directly through the model design (As introduced in Sec 2.1). First, we use view-aware positional embeddings to place all views into a shared geometric frame, allowing the model to understand how viewpoints relate spatially. Second, we introduce cross-view attention, where tokens from one view attend to tokens from all other views. This forces the model to align object positions and motions across viewpoints and prevents inconsistencies such as mismatched trajectories or geometry across views. These two mechanisms together structurally enforce multi-view coherence beyond simple joint prediction.
>
> - **W2**: "The approach requires few-shot fine-tuning for any new robotic embodiment. While this is a problem in general with VLA style policy networks, I would imagine that the video generation model, if trained on a variety of robotic embodiments, such as from Open-X-embodiments or similar datasets, in addition to the AgiBot world dataset, could help reduce this need for fine-tuning the GE-Base world model and require only fine-tuning the action decoder."
>
>    **A2**:Thank you for this insightful comment. We agree with the underlying idea: training in the visual space through a video generative modeling paradigm provides a natural way to share representations across different embodiments and avoids the difficulty of unifying heterogeneous action spaces. At the current stage, however, publicly available datasets vary significantly in quality, and high-quality dual-arm real-robot data, which is the primary focus of GE, remains scarce. Large collections such as Open-X-Embodiments include many demonstrations, but most of them are single-arm tasks and contain considerable noise, which makes them unsuitable for stable multi-embodiment pretraining. For this initial study, we therefore train GE-Base on AgiBot-World, a high-quality real-robot dataset with broad task and scene diversity. Even without cross-embodiment pretraining, the paper already shows that GE adapts well to new embodiments through full fine-tuning with only a small amount of additional data. We further conducted experiments showing that fine-tuning only the action decoder also yields meaningful gains; on the Agilex Cobot Magic Platform, one hour of demonstrations achieves a step success rate of about 0.5 on the cloth-folding task. Looking ahead, as higher-quality multi-embodiment datasets become available, such as the Galaxea Open-World Dataset, we plan to extend GE-Base with multi-embodiment video pretraining to further strengthen cross-embodiment generalization.

---

> ### Author Response · Authors · 2025-11-20
> **Response to Reviewer YQVX (2/3)**
>
> - **W3**: "Conceptually, the proposal of using video-based diffusion for learning latent spaces for robot manipulation has been explored previously. However, there is no mention of this in the paper, leave alone a comparison."
>
>   **A3**: Thank you for your valuable comment. We have added citations and discussion with prior video-diffusion-based manipulation methods in the related work sections of the revised version.
>   While these earlier approaches also employ video diffusion for action policy learning, our work introduces several conceptual and architectural advances:
>
>   1. First, most prior video diffusion–based approaches adopt a serial video-to-action pipeline, where the model must complete multi-step video denoising before extracting latent visual features, or in some cases even generate full pixel-space videos, for downstream action decoding. To maintain reasonable inference speed, these methods apply heavy latent compression, which inevitably discards the fine-grained motion and contact cues that precise manipulation relies on. In contrast, our framework introduces a parallel video–action decoder equipped with block-aligned multi-scale cross-attention that operates directly on full-resolution visual latents without any re-compression. This design preserves detailed spatial and interaction signals and leads to substantial improvements on fine-grained manipulation tasks such as box stacking and cloth folding.
>
>   3. Second, we introduce an efficient acceleration strategy for GE-Act based on our parallel architecture. Since GE-Act focuses on accurate action prediction rather than high-fidelity video generation, we adopt an asynchronous inference scheme: the heavy video diffusion decoder runs with only a single diffusion step, while the lightweight action decoder keeps multi-step decoding for precision. This setup preserves the required visual latent features while greatly improving speed. We also use a high–low frequency design in which the video decoder updates at a lower frequency and the action decoder at a higher one. Although this reduces temporal density, it retains essential spatial information and remains sufficient for manipulation. Experiments show that GE-Act Slow and GE-Act Fast reach similar accuracy, while GE-Act offers much higher throughput than previous video-based action policy models.
>
>   5. Third, conventional methods mostly rely on a single view. We introduce multi-view video generation to learn a cross-view latent space that aligns more closely with real robotic scenes and provides richer perceptual information.
>
>   7. Finally, we incorporate a sparse memory mechanism that further improves long-horizon task performance. These innovations collectively differentiate our approach from existing video-diffusion-based manipulation frameworks.
>
> - **W4**: "Notations in Sec. 2.1 are inconsistent with the use of bold font. The subscripts and superscripts aren't adequately explained."
>
>   **A4**: Thank you for the suggestion. We have revised the notation in Sec. 2.1 to ensure consistent use of bold fonts and provided clearer explanations for all subscripts and superscripts. These changes improve readability.
>
> - **Q1**: "Adding a mathematical explanation of how the losses are calculated."
>
>   **A5**: Thank you for the valuable comment. In the revised version, we have added the following mathematical formulations of the losses for both GE-Base and GE-Act.
>
>   The GE-Base is trained with a latent diffusion objective. Given VAE latents $\mathbf{l}$ of the target video chunk and a noisy latent $\tilde{\mathbf{l}} = (1-\sigma_\tau)\mathbf{l} + \sigma_\tau \boldsymbol{\epsilon}$, generated using Gaussian noise $\boldsymbol{\epsilon}\sim\mathcal{N}(0,I)$ at timestep $\tau$, the world model predicts the denoising velocity $\mathbf{v}_\theta$. Supervision is applied only to future frames via a conditioning mask $\mathbf{M}$, giving the training loss:
>
>   $$
>   \mathcal{L}\_{\text{video}} = w(\tau) \left\| \big(\mathbf{v}\_\theta - (\boldsymbol{\epsilon}-\mathbf{l})\big) \odot (1-\mathbf{M})\right\|_2^2
>   $$
>
>   Following the latent diffusion objective used in GE-Base, GE-Act trains its policy decoder using a noise-conditioned velocity-matching loss. Given ground-truth actions $\mathbf{u}$ and sampled noise $\boldsymbol{\epsilon}$ at diffusion timestep $\tau$, GE-Act predicts the denoising velocity $\mathbf{v}_\theta^{\text{act}}$. The supervision target is the diffusion velocity $\boldsymbol{\epsilon} - \mathbf{u}$, and the training objective is:
>
>   $$
>   \mathcal{L}\_{\text{act}} = w(\tau) \left\| \mathbf{v}\_\theta^{\text{act}} - (\boldsymbol{\epsilon} - \mathbf{u}) \right\|_2^2
>   $$
>
>   where $w(\tau)$ is the standard timestep weighting used in diffusion models. This objective mirrors the GE-Base training formulation and enables GE-Act to learn smooth and temporally coherent action trajectories directly in the latent space.

---

> ### Author Response · Authors · 2025-11-20
> **Response to Reviewer YQVX (3/3)**
>
> - **Q2**: "In Sec. 2.1, what is the difference between the next chunk $x_{1:N}^{(t)}$  and the sparse memory $\hat{x}\_{0:t-1}$."
>
>    **A6**: Thank you for the question. The next chunk $x_{1:N}^{(t)}$ is the $N$-frame video segment that the model predicts at step $t$ using the most recent generated history. In contrast, the sparse memory $\hat{x}_{0:t-1}$ is built by sparsely sampling frames from all previously generated chunks from step $0$ to $t-1$. This memory is not simply the previous chunk’s output; it is a long-range summary that provides stable conditioning, preserves global temporal consistency, and prevents the model from overfitting to short-term predictions.
>
>  - **Q3**: "In Lines 197-198, what is the difference between $\hat{x}\_t$ and $v\_t^{i}$? Isn't $v\_t^{(i)}$ a part of $\hat{x}\_t$? Shouldn't the model output be $\hat{x}\_{t+1}$ if the model predicts the next video chunk?"
>
>    **A7**: We apologize for the confusion. The notation $v_{\hat{t}}^{i}$ refers to the $i$-th view’s visual token extracted from the sparse memory, which is constructed by sampling frames from the previously generated chunks $0$ to $t-1$. It is not part of $\hat{x}_{t}$. We have revised the equations in the paper to clearly distinguish between information from step $t$ and step $t-1$, and updated the notation to avoid ambiguity.
>
>   - **Q4**: "In Figure 5, are the other methods trained/fine-tuned on robotic manipulation data as well? If not then, a fairer comparison is warranted, potentially by comparing against other robot world models as mentioned in the related works."
>
>     **A8**: Thank you for the question. After rechecking all baselines in Figure 5, we confirm that both general video generation models and world-model methods such as Cosmos were pretrained on substantial robotic manipulation video data. Cosmos, in particular, relies heavily on robot demonstrations during world-model training. Thus, the comparison in Figure 5 is already fair in terms of data exposure. We have also added qualitative comparisons with recent embodied video-generation baselines Video Predition Policy (VPP)[1] via text2video setting, which further show that GE-Base produces more coherent predictions.
>
>     | Model | Scene | Motion | Semantics | Score |
>     |-------|--------|---------|------------|----------|
>     | **GE-Act** | 0.9427 | 1.6676 | 2.0907 | 4.7010 |
>     | **VPP**    | 0.9251 | 0.2485 | 2.0050 | 3.1786 |
>
>
>   [1] Video Prediction Policy: A Generalist Robot Policy with Predictive Visual Representations. ICML 2025.

---

### Author Response · Authors · 2025-12-03
**General Comment to All Reviewers and the Area Chair**

We sincerely thank the Area Chair and reviewers for their constructive feedback and for recognizing the novelty and extensive real-world validation of our work. During the rebuttal, we actively addressed Reviewer XRta’s concerns by revising the Introduction and Related Work to better contextualize our method against concurrent video policies and clarifying technical details, which led to their satisfaction with our response. Additionally, we provided comprehensive clarifications to Reviewer tRPW regarding the unique advantages of our parallel architecture and asynchronous inference for real-time efficiency, while further elucidating the critical role of our memory mechanism in solving long-horizon tasks. We emphasize that Genie Envisioner establishes a unified, visual-centric world foundation platform trained on large-scale real-robot data, demonstrating superior performance in complex, fine-grained manipulation tasks where high-fidelity visual understanding is paramount, and we trust these revisions fully validate our contributions.

---

### Meta-Review · Area_Chair_JSf8 · 2026-01-08

**Summary:**

This paper proposes Genie Envisioner (GE), a large-scale vision-language-action framework for robotic manipulation that combines an instruction-conditioned video diffusion model (GE-Base) with a lightweight latent-space action decoder (GE-Act). Reviewers generally found the system technically sound, empirically strong, and well engineered, highlighting the scale of real-robot data, multi-view modeling, and the asynchronous video-action inference design as key strengths. Concerns centered on the paper’s positioning relative to prior video-action diffusion work, the clarity and scope of the “world model” claim, and the balance between real-robot evaluation and reproducible simulation benchmarks. Overall, while some conceptual and positioning issues remain, the work represents a substantial and practically relevant contribution to robotic manipulation.

**Reviewer Concerns:**

Addressed concerns

- Positioning with respect to prior work: The authors substantially expanded the related work to include GR-1/GR-2, VideoPolicy, UVA, Unified World Models, and other relevant joint video-action diffusion approaches. They clarified architectural distinctions, particularly the parallel video–action design, multi-view latent modeling, and asynchronous inference.

- Clarification of technical design choices: Multiple reviewers requested clearer explanations of multi-view consistency, sparse memory, asynchronous inference, and notation. These were addressed with revised descriptions, added mathematical formulations, and detailed rebuttal explanations.

- Empirical evaluation breadth: In response to concerns about limited simulation evaluation, the authors added results on established benchmarks (LIBERO-10 and RoboCasa) and provided additional ablations, including memory-module removal, demonstrating its impact—particularly for longer-horizon tasks.

- Efficiency and runtime: Comparisons to other efficient video-action diffusion models were added, showing that the asynchronous inference strategy enables real-time control despite the large backbone.

Outstanding concerns

- Terminology and conceptual framing of “world model”: While GE-Base is described as a world model, it primarily functions as an instruction-conditioned video prediction model whose outputs are consumed by a separate policy head. This differs from the classical action-conditioned dynamics models often associated with “world models” in robotics and RL. The terminology is defensible in the modern “video world model” sense, but remains somewhat overloaded and could be sharpened.

- Relative novelty versus concurrent methods: It is unclear that all claimed advantages over recent joint video-action diffusion approaches are fully demonstrated quantitatively, beyond the clearly novel memory mechanism and the asynchronous inference design.

**Reviewer Scores:**

- Reviewer YQVX: No change (8).
- Reviewer XRta: No change (8), explicitly stated they would keep their positive rating.
- Reviewer E6vL: Likely increase (4 → 6), as most clarity and positioning concerns were addressed and the reviewer indicated willingness to raise the score.
- Reviewer tRPW: No change (4). Despite additional experiments and discussion, core concerns about relative novelty and benchmark strength remain.

---

### Decision · Program_Chairs · 2026-01-26

Accept (Poster)